# Nanocompartment-confined polymerization in living systems

Yun Chen[1,2], Mengxuan Zuo[2], Yu Chen[3], Peiyuan Yu ✉[3], Xiaokai Chen[2], Xiaodong Zhang[2], Wei Yuan[2], Yinglong Wu[2], Wei Zhu[1] ✉ & Yanli Zhao[2] ✉

Polymerization in living systems has become an effective strategy to regulate cell functions and behavior. However, the requirement of high concentrations of monomers, the existence of complicated intracorporal interferences, and the demand for extra external stimulations hinder their further biological applications. Herein, a nanocompartment-confined strategy that provides a confined and secluded environment for monomer enrichment and isolation is developed to achieve high polymerization efficiency, reduce the interference from external environment, and realize broad-spectrum polymerizations in living systems. For exogenous photopolymerization, the light-mediated free-radical polymerization of sodium 4-styrenesulfonate induces a 2.7-fold increase in the reaction rate with the protection of a confined environment. For endogenous hydrogen peroxide-responsive polymerization, *p*-aminodiphenylamine hydrochloride embedded in a nanocompartment not only performs a 6.4-fold higher reaction rate than that of free monomers, but also activates an effective second near-infrared photoacoustic imaging-guided photothermal immunotherapy at tumor sites. This nanocompartment-confined strategy breaks the shackles of conventional polymerization, providing a universal platform for in vivo synthesis of polymers with diverse structures and functions.

Living organisms are replete with a wide variety of important chemical reactions in both intracellular and extracellular microenvironments, e.g., the biological synthesis of polysaccharides, proteins, and nucleic acids, to constitute the elementary components, confer essential functionalities, and modulate the biological process (Fig. 1a)[1–5]. Until now, various exogenous chemical reactions have been achieved in living systems for the synthesis of fluorescent probes or active drugs including click reaction, transfer hydrogenation, and removal of protective groups[6–8]. Polymerization reaction, a well-known method to produce multiple synthetic macromolecules with diversity and tunability properties, provides great possibilities for chemists to understand, analyze and regulate cell performance[9]. To date, some proof-of-

concept covalent and supramolecular polymerizations have been achieved in living cells to create new functional capabilities to modulate biological responses[1,10,11]. For illustration, Bradley et al. described the direct free radical photopolymerization inside living cells to manipulate cellular behavior[12]; Deisseroth and Bao et al. genetically engineered specific living neurons to synthesize electrical functional polymers to regulate the electrophysiological behaviors[13]; Weil et al. designed a platinum(II)-containing tripeptide that assembled into intracellular fibrillar nanostructures to disrupt energy homeostasis and cellular metabolism after response to endogenous hydrogen peroxide ($H_2O_2$)[14]. Despite these encouraging findings, artificial biological polymerization still encounters certain problems: (1) transportation of

[1]MOE International Joint Research Laboratory on Synthetic Biology and Medicines, School of Biology and Biological Engineering, South China University of Technology, Guangzhou 510006, P. R. China. [2]School of Chemistry, Chemical Engineering and Biotechnology, Nanyang Technological University, 21 Nanyang Link, Singapore 637371, Singapore. [3]Shenzhen Grubbs Institute and Department of Chemistry, Guangdong Provincial Key Laboratory of Catalysis, Southern University of Science and Technology, Shenzhen 518055, P. R. China. ✉e-mail: yupy@sustech.edu.cn; zhuwei86@scut.edu.cn; zhaoyanli@ntu.edu.sg

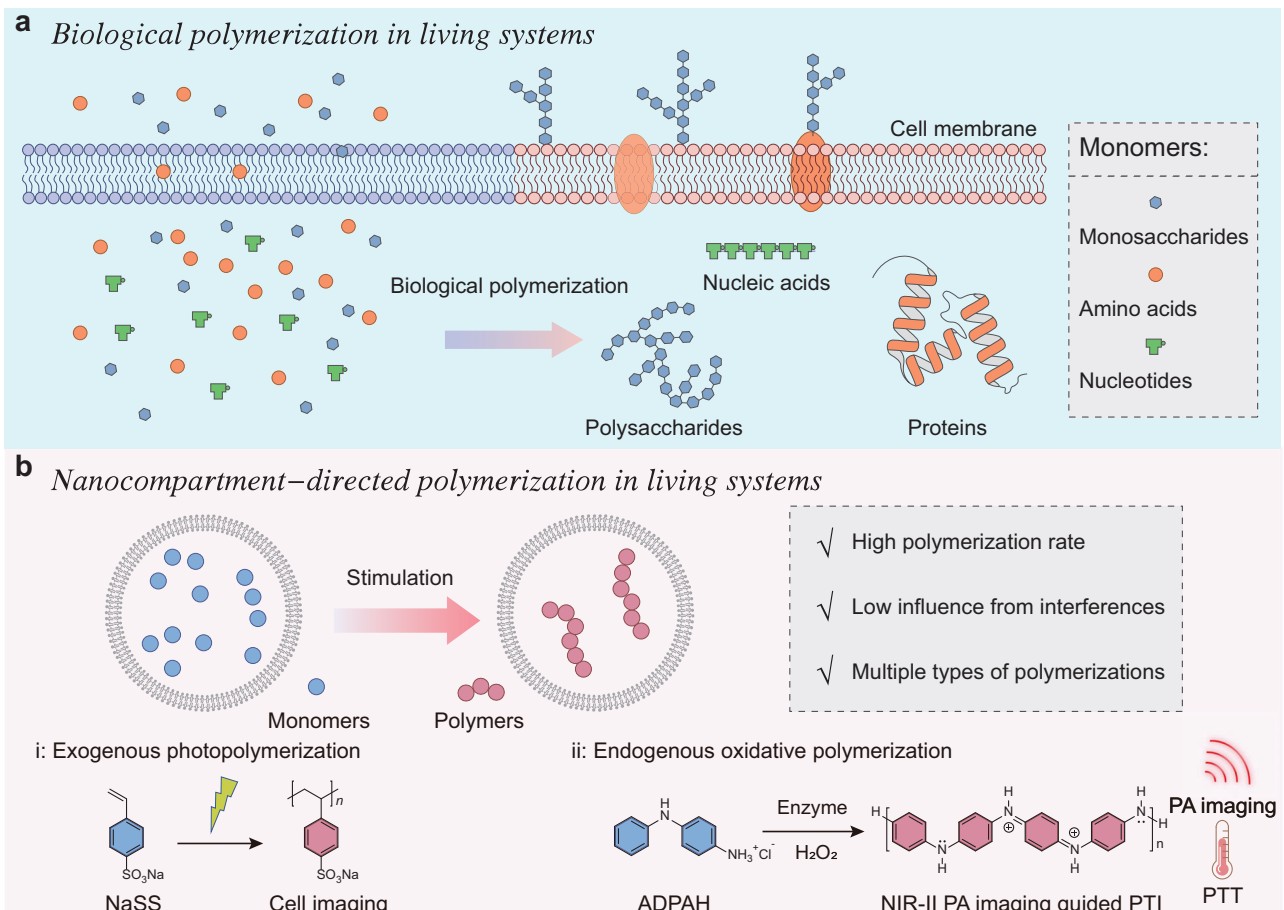

**Fig. 1 | Schematic diagram of polymerization in living systems. a** Biological polymerization of monosaccharides, amino acids, and nucleotides for the synthesis of polysaccharides, proteins, and nucleic acids to constitute the elementary components, confer essential functionalities, and modulate the biological process. **b** Nanocompartment-confined strategy to make a broad spectrum of polymerizations achievable in living systems: exogenous photopolymerization of the sodium salt of 4-styrenesulfonate (NaSS) and endogenous oxidative polymerization of *p*-aminodiphenylamine hydrochloride (ADPAH). NIR-II PA imaging second near-infrared photoacoustic imaging, PTI photothermal immunotherapy, PTT photothermal therapy.

high concentrations of monomers is always required; (2) the ubiquitous bioactive and metabolic substances in the living system might quench the polymerization process; (3) most polymerizations demand extra stimulates (ultraviolet (UV) irradiation, metal ions, etc.) to initiate the reaction; and (4) the synthetic polymers suffer from functional limitations to modulate additional biological responses[5]. The existence of these challenges hampers the pace of polymerization in living systems, thereby motivating researchers to actively seek solutions.

In this study, we develop a nanocompartment-confined strategy to successfully realize a broad spectrum of polymerizations in living systems (Fig. 1b). The nanocompartments provide a confined environment to ensure locally high concentrations of the monomers and increase the polymerization rate as the frequency of monomer collisions enhanced, which would in turn avoid the addition of high content of substrates, thereby decreasing their potential toxicity. Compartmentalization also leads to the creation of a secluded environment to reduce the influence of active substances on the polymerization process. This nanocompartment-confined strategy significantly enhances the feasibility of achieving artificial polymerizations in living systems. As examples, exogenous photopolymerization (e.g., light-mediated free-radical polymerization) and endogenous oxidative polymerization (e.g., $H_2O_2$-responsive polymerization) have been investigated (Fig. 1b). The experiment results showed that the nanocompartment-confined strategy could improve the reaction efficiency, reduce the influence of active substances on the polymerization process, and enable multiple polymerizations attainable in living systems. Moreover, the product of the

endogenous oxidative polymerization was able to harvest light to produce heat to ablate the tumor cells and activate effective photothermal immunotherapy (PTI). In this regard, the nanocompartment-confined effects induced a 3.5-fold temperature increase compared with the free monomers after ten-minute second near-infrared (NIR-II) light irradiation, illustrating the potential of dosage mitigation and toxicity reduction. Overall, this nanocompartment-confined polymerization breaks the borderlines of polymerization in living systems and enables the construction of polymers with diverse structures and functions in vivo, providing a universal platform for the investigation of polymerization in living organisms.

## Results

### Nanocompartment-directed confined exogenous photopolymerization

Structure-directing nanocompartments, e.g., emulsions, micelles, and vesicles, have proven to be a powerful tool to localize the reaction process in their vicinity for steering and controlling the polymerization reaction[15]. Here a vesicular compartment fabricated by Aerosol-OT (AOT, sodium bis(2-ethyl-1-hexyl) sulfosuccinate), FDA-approved anionic oil-soluble surfactant, was exploited to investigate the vesicle-directed confined polymerization (Fig. 2a). AOT vesicles were formed by solvent injection method and characterized by transmission electron microscopy (TEM), which reflected a sunken surface and vesicular structure with a diameter of ~80 nm (Fig. 2b). As one of the most common polymerization techniques, light-mediated free radical

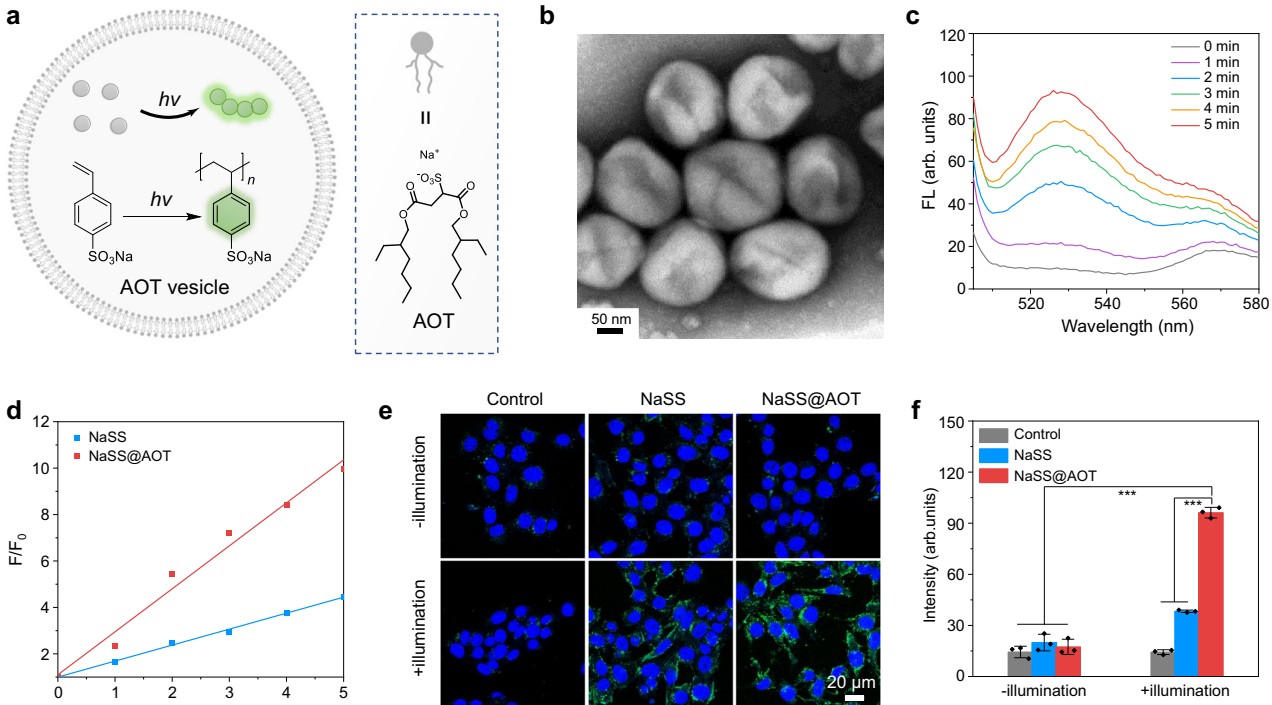

**Fig. 2 | Nanocompartment-directed confined exogenous photopolymerization.**
**a** Schematic illustration of AOT-directed photopolymerization of NaSS. **b** TEM characterization of AOT vesicle. The experiment was repeated three times with similar results. **c** Emission spectrum ($\lambda_{ex}$ = 480 nm) of the solution of NaSS@AOT before and after UV illumination (365 nm, 5 min). **d** Reaction rate of NaSS and NaSS@AOT with UV illumination (365 nm, 5 min). $F/F_0$ represents the ratio of fluorescence signals of poly(NaSS) after and before UV illumination (365 nm, 5 min). **e** CLSM images of 4T1 cells following various treatments (n = 3). Blue fluorescent signal was from Hoechst 33342 cell nucleus staining and green fluorescent signal indicated the formation of poly(NaSS), respectively. **f** Mean fluorescence intensity (MFI) of polymerized NaSS level after various treatments (n = 3). NaSS@AOT with illumination versus other groups: $p < 0.001$. Data were presented as mean ± standard deviation (SD). Statistical significance in (**f**) was calculated via one-way analysis of variance (ANOVA) followed by Tukey post-hoc test. ***$p < 0.001$.

polymerization has been widely employed as a representative exogenous example for investigation. Photopolymerization of the sodium salt of 4-styrenesulfonate (NaSS) with the assistance of photoinitiator Irgacure 2959 displayed a gradual enhancement of the fluorescence intensity at 528 nm due to the generation of fluorescent poly(NaSS) (Supplementary Fig. 1)[12]. Thereafter, NaSS and Irgacure 2959 were enriched in the AOT vesicles, offering the advantages of locally high concentration for quick polymerization. The rate of increase in fluorescence intensity, measured from the slope of the intensity, was improved by 2.7-fold compared with the free monomers at the same conditions (Fig. 2c, d). Moreover, the intracellular polymerization of NaSS@AOT was conducted. The confocal laser scanning microscopy (CLSM) images revealed no evident fluorescent signals in the NaSS and NaSS@AOT treated 4T1 cells (Fig. 2e). Impressively, after five-minute 365 nm light illumination (2.5 mW cm⁻²), fluorescence enhancement for the NaSS@AOT group (5.5-fold) was 2.9 times higher than that of NaSS group (1.9-fold), validating the nanocompartment-confined effects (Fig. 2f). Noted that NaSS and Irgacure 2959 displayed little influence on the viability of mouse breast 4T1 cancer cells with the concentration up to 25 mM and 1.6 mM, respectively (Supplementary Fig. 2). In addition, the UV illumination (5 min at 365 nm) also showed negligible adverse effects on cell viability (Supplementary Fig. 3).

## Nanocompartment-directed confined endogenous oxidative polymerization

Until now, most implemented biological polymerization always requires additional external stimulation, e.g., metal ions, irradiation, or heat, to the intravital synthesis of non-naturally occurring polymers, which suffers from more restrictions for practical applications. Potentially, endogenous signals (e.g., $H_2O_2$, low pH, and enzymes)

represent appealing stimuli to trigger biological polymerization[16]. Horseradish peroxidase (HRP), an enzyme known to catalyze $H_2O_2$-dependent one-electron oxidations, could promote the polymerization of natural or unnatural aromatic monomers to obtain corresponding polyaromatic products[15]. Previous research has demonstrated the effectiveness of AOT anionic vesicles in facilitating HRP-catalyzed reactions, particularly when utilizing aniline analogues as the monomers[17]. Here, aromatic amine monomers (e.g., o-phenylenediamine (OPD), p-phenylenediamine (PPD), m-phenylenediamine (MPD), p-aminodiphenylamine (ADPA), 4,4′-iminodianiline (NDA), and ADPA in the form of amine hydrochloride salt (ADPAH)) was separately entrapped in the cavity of the AOT vesicle, and $H_2O_2$ was utilized to trigger the oxidative polymerization as it could transport through the bilayers due to the low polarity (Fig. 3a)[18]. After the addition of $H_2O_2$, the polymerization occurred as revealed by the changed ultraviolet/visible/near–infrared absorption spectra (UV–vis/NIR, Fig. 3b, c). For these enzymatic reactions, their initial velocity was disclosed by the linear change of absorbance over time (Supplementary Figs. 4 and 5). As summarized in Supplementary Fig. 6, the polymerization rate was significantly increased in substrates containing AOT vesicles compared to those with free monomers due to the localized high concentration of monomers within the confined nanocavity. For instance, ADPAH@AOT demonstrated a polymerization rate of 6.4 ± 0.4 times higher than that of the ADPAH group. In addition to their role in molecular enrichment, nanocompartments also facilitated molecular isolation, reducing the influence of active substances on the polymerization process. As depicted in Fig. 3d, the introduction of fresh cell supernatant, which acted as an influencing factor, had a more significant impact on monomers without vesicles. Specifically, with the interference, ADPAH@AOT retained 27 ± 7.1% of its original activity, while

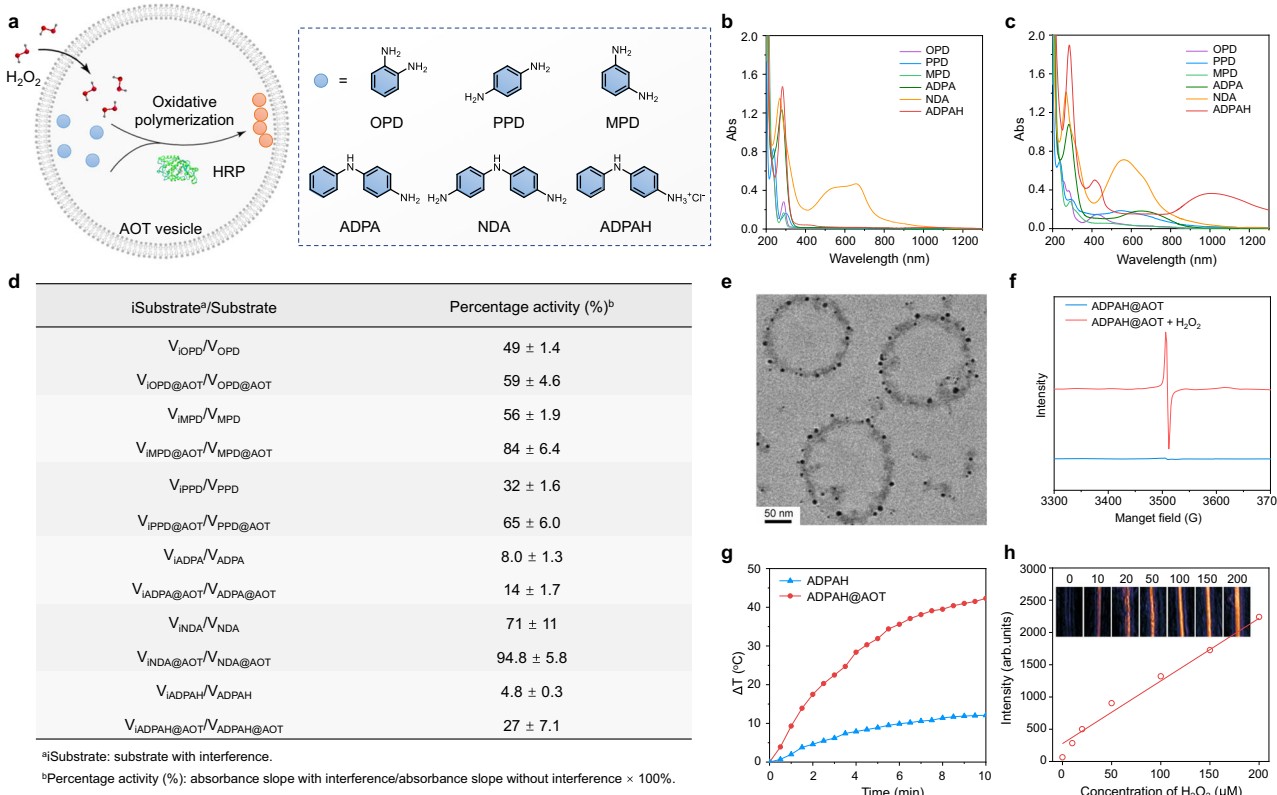

**Fig. 3 | Nanocompartment-directed confined endogenous oxidative polymerization. a** Schematic illustration of AOT-directed oxidative polymerization of aniline analogues. **b, c** Absorption spectra of OPD, PPD, MPD, ADPA, NDA and ADPAH (**b**) before and (**c**) after response to $H_2O_2$ (100 μM). **d** Table summaries of the initial velocity ratio of the enzymatic reactions with or without AOT nano-compartments before and after the addition of fresh cell supernatant to influence the polymerization process. **e** TEM characterization of ADPAH@AOT vesicle after

response to $H_2O_2$ (100 μM). The experiment was repeated three times with similar results. **f** ESR spectrum of ADPAH@AOT before and after response to $H_2O_2$ (100 μM). **g** Photothermal heating curves of ADPAH and ADPAH@AOT solution after the addition of $H_2O_2$ (100 μM) under 1064 nm photoirradiation at 1 W cm$^{-2}$. **h** PA images and intensities of ADPAH@AOT with various concentrations of $H_2O_2$ (0, 10, 20, 50, 100, 150, and 200 μM) under 1064 nm laser irradiation.

the nanocompartment-deficient ADPAH group only reserved 4.8 ± 0.3% of its activity. The aforementioned findings validate that the presence of a confined and secluded environment has a beneficial impact on enhancing reaction efficiency and establishing a stable space for polymerization.

Among these substrates, ADPAH@AOT revealed an obvious NIR-II absorption with a peak at ~1003 nm upon exposure to $H_2O_2$ (Fig. 3c). Noted that the observation aligns with earlier pioneer research that utilized AOT to assist in the polymerization of ADPA and demonstrated the successful generation of polyaniline in its emeraldine salt form (PANI-ES) with absorption in NIR-II region[13,15,17]. Briefly, HRP catalyzed $H_2O_2$-dependent one-electron oxidation with ADPAH leads to the formation of the diimine, which was unstable in aqueous solution and underwent a coupling reaction with each other to form Poly-ADPAH (PADPAH) (Supplementary Fig. 7)[19,20]. Considering the sulfonate-rich structure of AOT, the anionic interfaces contributed to doping with PADPAH to form a diradical dication structure with NIR-II absorbance. TEM image indicated that there were various small black nanoparticles generated near the vesicle membrane after responding to $H_2O_2$, verifying the interaction of PADPAH with the vesicle templates (Fig. 3e). To clarify the existence of radical dication, electron spin resonance (ESR) spectrum of ADPAH@AOT before and after polymerization was studied. The appearance of an EPR signal after polymerization indicated the existence of unpaired electrons in the polaron state of PADPAH (Fig. 3f). The highest occupied molecular orbital (HOMO)-the lowest unoccupied molecular orbitals (LUMO) gap of the dimeric ADPAH were calculated to be 6.47 eV, higher than that of its diradical dication product (3.90 eV), confirming the red-shifted absorbance of

dimeric ADPAH in the presence of AOT vesicle (Supplementary Fig. 8). Further theoretical analysis of diradical dication of PADPAH in Supplementary Fig. 9 predicted a strong absorbance peak at ~981 nm, which coincide well with the UV/vis−NIR measurement. The heterogeneous reactive sites of the other four analogues hindered the formation of a large π−conjugation system, resulting in the unsuccessful formation of long-wavelength absorption products, even in their hydrochloride forms (Supplementary Fig. 10).

The overexpressed $H_2O_2$ in tumor tissues makes ADPAH@AOT an attractive candidate for activating NIR-II photothermal therapy (PTT). In the NIR-II window, the combination of low absorption by main endogenous absorbers and the reduced scattering from various tissues facilitates numerous advantages, including deep tissue penetration capabilities (at least 20 mm), high maximum permission energy, and excellent sensitivity[21–24]. From this perspective, after NIR-II laser (1064 nm) irradiation, the temperature of ADPAH@AOT with the addition of $H_2O_2$ (100 μM) quickly increased 42.3 °C in 10 min, 3.5 times higher than that of free ADPAH monomer at the same condition (Fig. 3g). From the UV/vis−NIR spectra, this oxidative polymerization was triggered by $H_2O_2$ in a time- and concentration-dependent manner, and it also showed specificity towards $H_2O_2$ (Supplementary Fig. 11). The further investigation of ADPAH concentration-, laser-density-, and $H_2O_2$ concentration-dependent temperature change of ADPAH@AOT in response to 1064 nm laser was studied and compared in Supplementary Fig. 12. The polymerized ADPAH@AOT showed high photothermal stability after 4 times heating and cooling cycle (Supplementary Fig. 13). As the photothermal effect could generate detectable acoustic waves that can be converted into imaging signals,

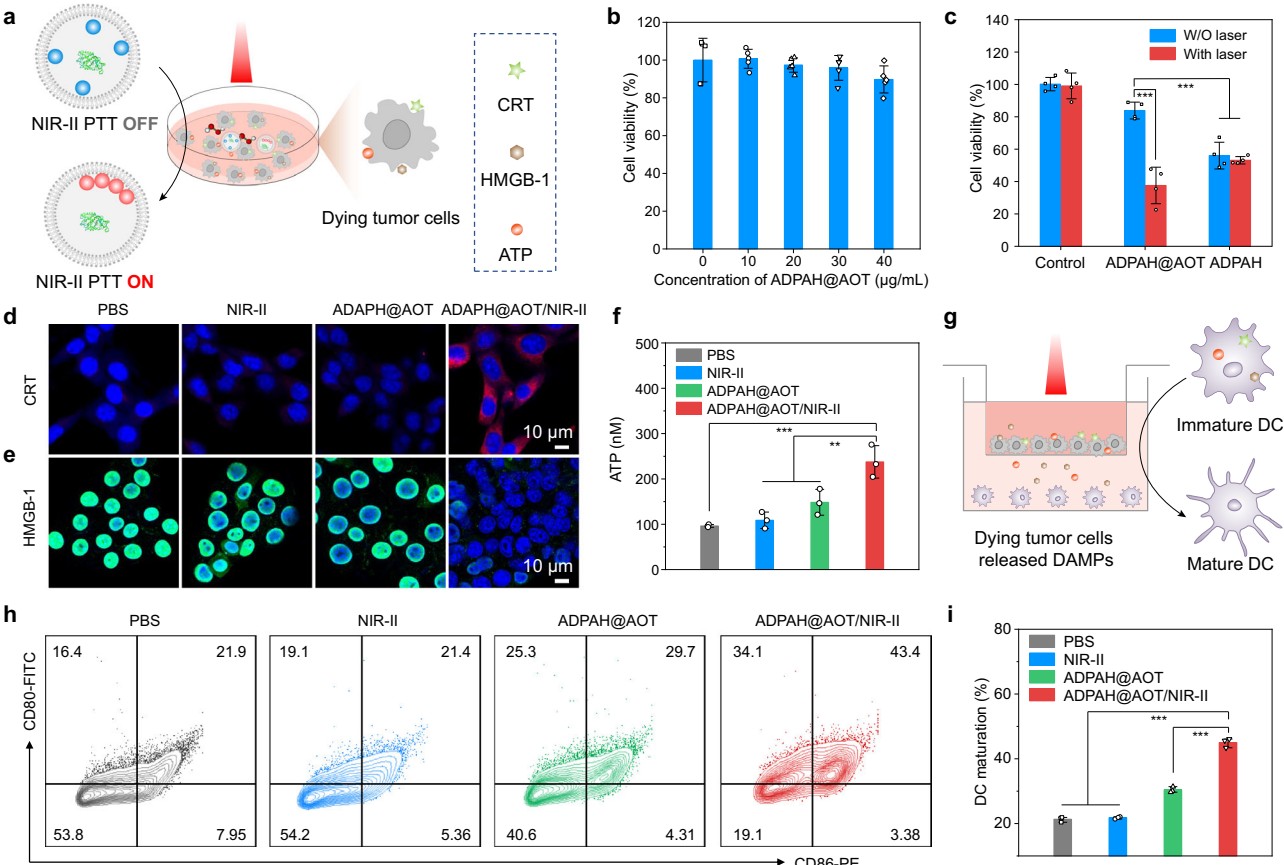

**Fig. 4 | Nanocompartment-directed confined endogenous oxidative polymerization in living cells. a** Schematic illustration of NIR-II laser irradiation of ADPAH@AOT induced immunogenic cell death. **b** Relative cell viabilities of NIH-3T3 cells after treatment with ADPAH@AOT at different concentrations ($n = 5$). **c** Relative cell viabilities of 4T1 cells after treatment with ADPAH@AOT and ADPAH with or without 1064 nm irradiation for 5 min (1 W cm$^{-2}$) ($n = 4$). ADPAH@AOT with laser versus ADPAH@AOT without laser: $p < 0.001$; ADPAH@AOT without laser versus ADPAH with or without laser: $p < 0.001$. **d, e** Immunofluorescence imaging of (**d**) CRT and (**e**) HMGB-1 in 4T1 tumor cells after treatment with ADPAH@AOT with and without 1064 nm laser irradiation at 1 W cm$^{-2}$ ($n = 3$). **f** ATP content of 4T1 cells supernatant after treatment with ADPAH@AOT with and without 1064 nm laser irradiation at 1 W cm$^{-2}$ ($n = 3$). ADPAH@AOT/NIR-II versus ADPAH@AOT: $p = 0.009$; ADPAH@AOT/NIR-II versus NIR-II: $p = 0.001$; ADPAH@AOT/NIR-II versus PBS: $p < 0.001$. **g** Schematic illustration for DC maturation using a transwell system. **h** Flow cytometry plots and **i** quantitative analysis of DC maturation (CD80$^+$CD86$^+$) after different treatments ($n = 3$). ADPAH@AOT/NIR-II versus other groups: $p < 0.001$. Data were presented as mean ± SD. Statistical significance in (**c**), (**f**) and (**i**) was calculated via one-way ANOVA followed by Tukey post-hoc test. **$p < 0.01$, and ***$p < 0.001$.

the possibility of using photoacoustic (PA) imaging for the H$_2$O$_2$-triggered polymerization was further examined[25]. With the increase of H$_2$O$_2$ concentrations, the observed PA signals of ADPAH@AOT at 1064 nm showed a significant increase, consistent with the H$_2$O$_2$-dependent absorption increase (Fig. 3h).

### Nanocompartment-directed endogenous oxidative polymerization in living cells

To evaluate the nanocompartment-directed oxidative polymerization towards living cells (Fig. 4a), their biocompatibility towards NIH-3T3 fibroblasts was first evaluated. As shown in Fig. 4b, ADPAH@AOT displayed negligible cytotoxicity against NIH-3T3 cells with or without NIR-II irradiation, indicating the high biocompatibility of ADPAH@AOT towards normal cells (Supplementary Fig. 14). Subsequently, 4T1 cells were treated with ADPAH@AOT in the presence of exogenous 100 × 10$^{-6}$ M H$_2$O$_2$. As shown in Fig. 4c, the treatment of 20 μg mL$^{-1}$ of ADPAH@AOT caused ~62 ± 11% cell death in comparison with the control group after being irradiated by 1064 nm light for 5 min at the power density of 1 W cm$^{-2}$. However, for the free ADPAH group, negligible toxicity increase was observed due to the low-temperature increase after irradiation. Notably, ADPAH exhibited higher cytotoxicity than that of ADPAH@AOT, suggesting that the nanocompartment could reduce the potential cytotoxicity of free monomers. Note that

100 × 10$^{-6}$ M H$_2$O$_2$ has little influence on the 4T1 cell viability (Supplementary Fig. 15).

NIR-II PTT is a promising immune therapy approach to activate immunogenic cell death by the exposure of damage-associated molecular patterns (DAMPs), e.g., calreticulin (CRT), high mobility group box 1 (HMGB-1), and adenosine triphosphate (ATP)[26,27]. To evaluate the activation of immunogenic cell death after irradiation of the synthetic polymer, 4T1 cells were pre-treated with H$_2$O$_2$ (100 μM) and followed by treatment of ADPAH@AOT. An upregulated CRT expression was shown in the CLSM images of ADPAH@AOT/NIR-II group, while negligible red signals were observed in the PBS group (Fig. 4d). The signal of HMGB-1 was located predominantly in the nuclei in the PBS group, whereas it was significantly released in the ADPAH@AOT/NIR-II group (Fig. 4e). Moreover, the extracellular ATP levels in ADPAH@AOT/NIR-II were 1.6- and 2.5-fold higher than that of ADPAH@AOT and PBS group, respectively (Fig. 4f).

Afterward, a Transwell system was adopted to explore the DAMPs promoted dendritic cell (DC) maturation. 4T1 cells in the upper chambers were pre-treated with ADPAH@AOT in the presence of H$_2$O$_2$ (100 μM) and followed by a NIR-II laser irradiation, then the immature bone marrow-derived dendritic cells (BMDCs) were seeded in the lower chambers (Fig. 4g). In contrast to the ADPAH@AOT group,

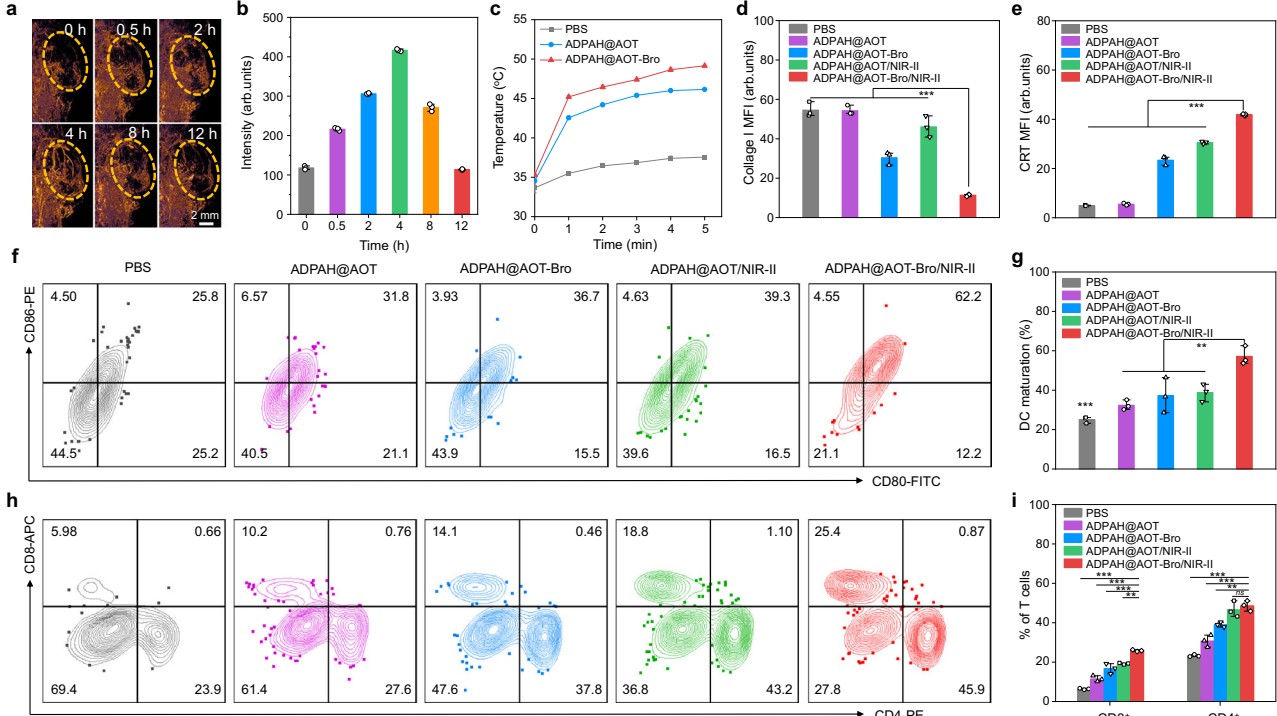

**Fig. 5 | Nanocompartment-directed confined endogenous polymerization in living organisms. a** In vivo PA imaging and **b** corresponding PA signal intensity of the 4T1-bearing tumor mice after intertumoral administration of ADPAH@AOT-Bro (1064 nm, $n = 3$). **c** Mean surface temperature of corresponding 4T1-tumor-bearing mice under 1064 nm irradiation (1 W cm$^{-2}$) at 4 h post intertumoral injection of PBS, ADPAH@AOT, and ADPAH@AOT-Bro ($n = 3$). **d** Quantitative analysis of collage I from the immunofluorescence staining of primary tumor sections following various treatments ($n = 3$). ADPAH-Bro@AOT/NIR-II versus other groups: $p < 0.001$. **e** Quantitative analysis of CRT from the immunofluorescence staining of primary tumor sections following various treatments ($n = 3$). ADPAH-Bro@AOT/NIR-II versus other groups: $p < 0.001$. **f** Representative flow cytometric plots and **g** quantification of spleen-derived mature DCs (CD80$^+$CD86$^+$) following various

treatments ($n = 3$). ADPAH-Bro@AOT/NIR-II versus ADPAH@AOT/NIR-II: $p = 0.009$; ADPAH-Bro@AOT/NIR-II versus ADPAH@AOT-Bro: $p = 0.005$; ADPAH-Bro@AOT/ NIR-II versus ADPAH@AOT: $p = 0.001$; ADPAH-Bro@AOT/NIR-II versus PBS: $p < 0.001$. **h** Representative flow cytometric plots and **i** quantification of spleen-infiltrating T lymphocytes (CD8$^+$ and CD4$^+$) following various treatments ($n = 3$). $p$ value for CD8$^+$ T cells: ADPAH-Bro@AOT/NIR-II versus ADPAH@AOT/NIR-II: $p = 0.001$; ADPAH-Bro@AOT/NIR-II versus three other groups: $p < 0.001$. $p$ value for CD4$^+$ T cells: ADPAH-Bro@AOT/NIR-II versus ADPAH@AOT/NIR-II: $p = 0.886$; ADPAH-Bro@AOT/NIR-II versus ADPAH@AOT-Bro: $p = 0.009$; ADPAH-Bro@AOT/ NIR-II versus ADPAH@AOT and PBS: $p < 0.001$. Data were presented as mean ± SD. Statistical significance in (**d**), (**e**), (**g**), and (**i**) was calculated via one-way ANOVA followed by Tukey post-hoc test. ns not significant, **$p < 0.01$, and ***$p < 0.001$.

where only a weak maturation rate was observed (30 ± 0.9%), the photoirradiated ADPAH@AOT significantly increased the maturation of DCs to 45 ± 1.4% (Fig. 4h, i and Supplementary Fig. 16). Besides, the highest expression level of interleukin 6 in the culture medium was monitored in ADPAH@AOT/NIR-II group (Supplementary Fig. 17). These results verified the activable NIR-II PTT could trigger DAMPs release, promote DC maturation, and show the potential in promoting T cells infiltration.

## Nanocompartment-directed confined endogenous oxidative polymerization in living organisms

To evaluate this nanocompartment-directed oxidative polymerization in living systems, NIR-II PA imaging was first utilized to visualize the process. As shown in Fig. 5a, b, obvious PA signals were observed after just 0.5 h intertumoral treatment of ADPAH@AOT, and the signals quickly increased over time. The incremental PA signals originated from the tumor-overexpressed H$_2$O$_2$-triggered polymerization, indicating successful oxidative polymerization in living system. This endogenous oxidative polymerization also contributed to activatable PTT as revealed by the increased temperature (11.6 ± 0.1 °C) in ADPAH@AOT treated tumors after NIR-II irradiation compared with a slight temperature increase (3.9 ± 0.1 °C) in PBS group after 5 min of irradiation (Fig. 5c and Supplementary Fig. 18).

However, despite PTT having been demonstrated to elicit immunogenic cell death and promote DC maturation, their immunological antitumor efficacy was often restrained by low infiltration of T cells in the tumor sites[28,29]. Tumor extracellular matrix (ECM) provides a collagen network and serves as a natural barrier for immune evasion and attenuation of antitumor therapy[30]. Bromelain, a protease, not only possesses anti-inflammatory, anticancer, and immunomodulatory activities, but also exhibits a preferred catalytic temperature of ~45 °C[31,32]. This temperature preference makes it suitable for a photothermal-mediated enzyme switch in efficient collagen digestion[33]. In this regard, bromelain was incorporated in ADPAH@AOT to promote ECM degradation, assist antitumor activity, facilitate immunomodulatory, and enhance T cells infiltration into the tumor tissues. The collage degradation efficacy of ADPAH@AOT-Bro/ NIR-II group, expressed as gelatin digestion units (GDU), was calculated to be 3253.6 ± 169.8 GDU g$^{-1}$, higher than that without NIR-II laser irradiation (1749.6 ± 141.8 GDU g$^{-1}$, Supplementary Fig. 19). These results confirmed that the enzymatic activity of ADPAH@AOT-Bro could be triggered by the photothermal heating, as the solution temperature increased circa 24.8 °C under NIR-II laser irradiation (Supplementary Fig. 20). To our delight, the temperature in ADPAH@AOT-Bro treated tumors exceeded ADPAH@AOT group and reached 49.2 °C after NIR-II irradiation (Fig. 5c and Supplementary Fig. 18). The promoted temperature increase may be attributed to the enhanced accumulation of ADPAH@AOT-Bro in tumor sites after collage digestion. The collagen levels in the xenografted tumors were studied by immunofluorescence staining of the tumor slices (Supplementary

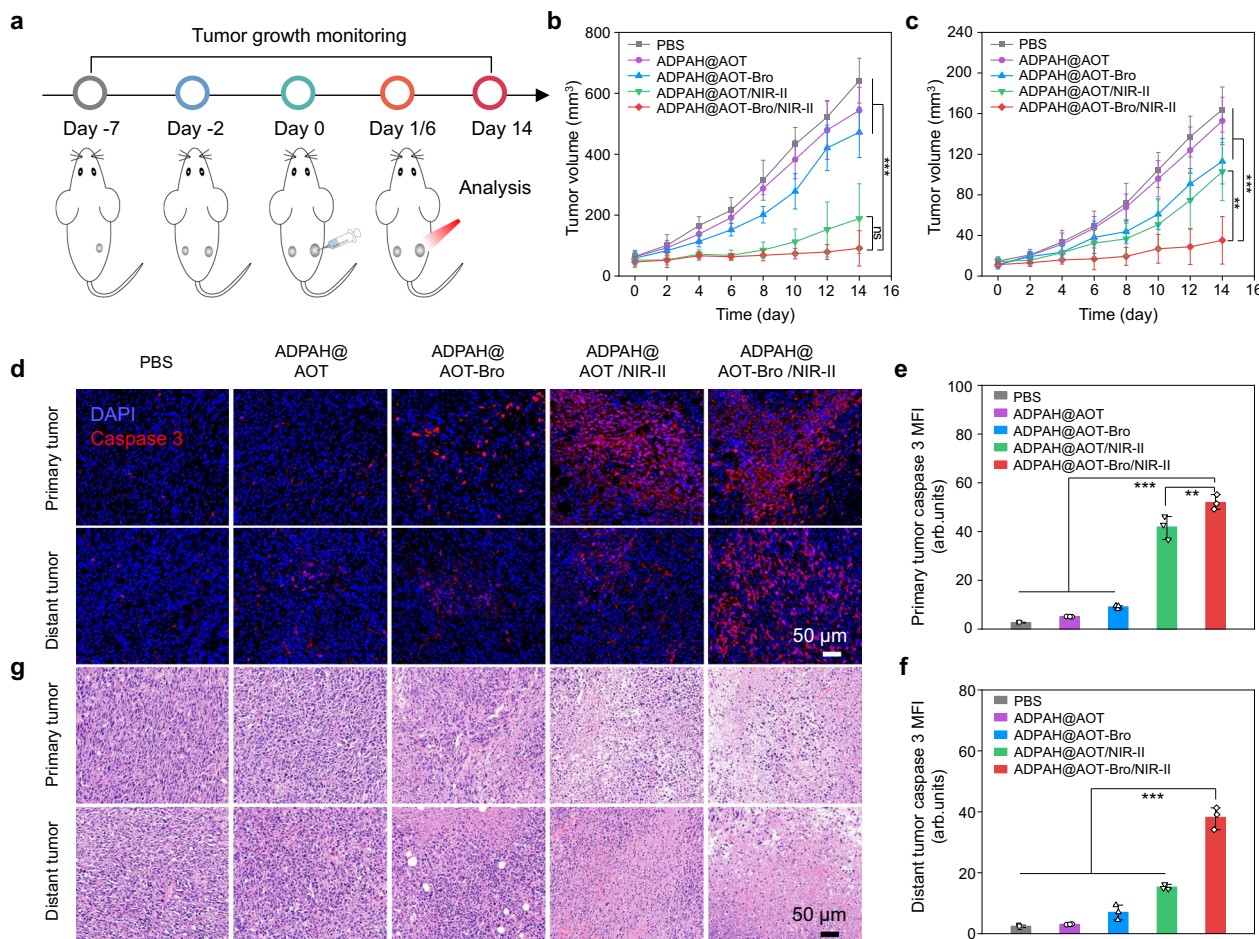

**Fig. 6 | In vivo antitumor performance of ADPAH@AOT-Bro. a** Schematic illustration of in vivo antitumor schedule. **b, c** Tumor growth curves of (**b**) primary and (**c**) distant tumors in 4T1-tumor-bearing mice following various treatments (1064 nm, 5 min, $n = 5$). ADPAH-Bro@AOT/NIR-II versus ADPAH@AOT/NIR-II in (**b**): $p = 0.373$; ADPAH-Bro@AOT/NIR-II versus other three groups in (**b**): $p < 0.001$; ADPAH-Bro@AOT/NIR-II versus ADPAH@AOT/NIR-II in (**c**): $p = 0.002$. ADPAH-Bro@AOT/NIR-II versus other three groups in (**c**): $p < 0.001$. **d** Immunofluorescent images of active caspase-3 (red) in primary and distant tumor sections following various treatments. **e, f** Quantification of expression level of active caspase-3 in (**e**) primary and (**f**) distant tumors sections from euthanized mice following various treatments ($n = 3$). ADPAH-Bro@AOT/NIR-II versus ADPAH@AOT/NIR-II in (**e**): $p = 0.005$. ADPAH-Bro@AOT/NIR-II versus other three groups in (**e**): $p < 0.001$; ADPAH-Bro@AOT/NIR-II versus other groups in (**f**): $p < 0.001$. **g** H&E-stained images of the primary and abscopal tumor sections from euthanized mice following various treatments ($n = 3$). Data were presented as mean ± SD. Statistical significance in (**b**), (**c**), (**e**) and (**f**) was calculated via one-way ANOVA followed by Tukey post-hoc test. ns not significant, **$p < 0.01$, and ***$p < 0.001$.

Fig. 21). Significant red fluorescence corresponding to the level of collagen I was observed in the control and ADPAH@AOT group. The assistance of bromelain declined the red fluorescence by 1.8-fold as compared with the ADPAH@AOT group, and a further ~3.0-fold fluorescence reduction was detected after NIR-II irradiation, indicating the ascendance of enzyme activities with the help of photothermal effect (Fig. 5d).

To reveal the endogenous polymerization contributed to NIR-II PTI, a model of BALB/c mice bearing bilateral 4T1 tumors was developed. Firstly, the in vivo immunogenic cell death activation was investigated by immunofluorescence staining of CRT expression in the primary tumor sections, and a strong red fluorescence was observed in the irradiated ADPAH@AOT group, which was 5.5- and 6.2-fold relative to those of ADPAH@AOT and PBS group, respectively (Fig. 5e and Supplementary Fig. 22). Moreover, ADPAH@AOT with NIR-II irradiation also triggered a significantly increased population of mature DCs (CD80$^+$CD86$^+$, 39 ± 4.5%), compared to that of ADPAH (32 ± 2.5%) and PBS (25 ± 1.6%) group, respectively (Fig. 5f, g and Supplementary Figs. 23 and 24). After the combination of bromelain, the CRT exposure and DC maturation were enhanced. Especially, the photo-irradiated ADPAH@AOT-Bro group (ADPAH@AOT-Bro/NIR-II) exhibited the CRT

exposure and DC maturation levels 1.4 and 1.5 times higher than those of ADPAH@AOT/NIR-II group, respectively. The matured DCs and disturbed ECMs would further promote the intratumoral infiltration of cytotoxic CD8$^+$ T lymphocytes and CD4$^+$ helper T cells. Photoirradiated ADPAH@AOT-Bro group induced the highest infiltration levels of T lymphocytes in both the primary and distant tumors (Supplementary Fig. 25). In detail, the population of CD8$^+$ T cells in ADPAH@AOT-Bro/NIR-II group reached the highest expression levels, which was 2.0- and 1.5-fold higher than ADPAH@AOT or ADPAH@AOT-Bro group in the primary tumor, and 2.1- and 1.7-fold higher in the distant tumor, respectively (Supplementary Figs. 26 and 27). Concurrently, in the ADPAH@AOT-Bro/NIR-II group, the percentages of CD8$^+$ T cells and CD4$^+$ T cells in the spleen were 4.1- and 2.1-fold higher relative to that of the PBS group, respectively (Fig. 5h, i). Furthermore, ADPAH@AOT-Bro/NIR-II group also triggered the highest immunostimulatory cytokines secretion (Supplementary Fig. 28). For instance, interferon γ (IFN-γ) levels in ADPAH@AOT-Bro/NIR-II treated mice were 148.2 ± 39.4 pg mL$^{-1}$, higher than that of ADPAH@AOT-Bro group (101.5 ± 30.6 pg mL$^{-1}$) and ADPAH@AOT group (82.6 ± 10.3 pg mL$^{-1}$), respectively. These data implied that the H$_2$O$_2$-responsive polymerization activated effective NIR-II PTT at tumor sites, which could

also activate immunogenic cell death, promote DC maturation, and assist with ECM degradation bromelain to facilitate the T cell infiltration (Supplementary Fig. 29).

Encouraged by the satisfactory immunological effect, the antitumor performance was then investigated using a model of BALB/c mice bearing bilateral 4T1 tumors (Fig. 6a). The growth of both tumors was monitored every other day. At the end of the treatment, ADPAH@AOT-Bro/NIR-II group exhibited the best antitumor efficacy with an average tumor inhibitory rate of 86%, 1.2 and 3.2 times higher than that of ADPAH@AOT/NIR-II and ADPAH@AOT-Bro groups, respectively (Fig. 6b). For the distant tumors, the most retarded growth was seen in ADPAH@AOT-Bro/NIR-II group with 79% tumor inhibition, 2.1 and 2.5 times higher than that of ADPAH@AOT/NIR-II and ADPAH@AOT-Bro groups, respectively (Fig. 6c).

The therapeutic outcomes were further verified by immunofluorescence and hematoxylin and eosin (H&E) staining of both the primary and abscopal tumors. The fluorescence intensity of CY3-labeled anticaspase-3 antibody, representing the degree of cell apoptosis, was higher in ADPAH@AOT-Bro/NIR-II group than that in other groups on both flank sides (Fig. 6d). In detail, the MFI of caspase-3 in ADPAH@AOT-Bro/NIR-II group was 1.2-, 5.7-, 10.2-, and 19.7-fold higher than that of ADPAH@AOT/NIR-II, ADPAH@AOT-Bro, ADPAH@AOT, and PBS group in primary tumors, and 2.5-, 5.5-, 12.5-, and 15.7-fold higher than that in abscopal tumors, respectively (Fig. 6e, f). H&E results displayed the most obvious damage in ADPAH@AOT-Bro/NIR-II treated tumor sections (Fig. 6g). In the course of treatment, the body weight remained stable for the treatment groups (Supplementary Fig. 30) and no detectable damage was found in the H&E staining of the major organs in mice after various treatments (Supplementary Fig. 31), indicating the high biosafety and good biocompatibility. These results indicated that the nanocompartment-confined strategy was capable of realizing in vivo polymerization and this endogenous oxidative polymerization was able to activate effective PTI and achieve strong antitumor effects.

## Discussion

In summary, a nanocompartment-confined strategy was established to achieve various types of polymerizations in living systems. This compartmentalization provides a confined environment for monomers enrichment and isolation, offering the possibility to increase the polymerization rate and decrease the potential toxicity. The commonly used exogenous light-mediated free-radical polymerization and endogenous $H_2O_2$-responsive polymerization were selected to study nanocompartment-directed biological polymerization. The illumination of NaSS in AOT vesicle induced a 2.7-fold increase in the rate of polymerization and a 2.9-fold enhancement of fluorescence in living cells than those of NaSS group, respectively.

The nanocompartment-directed oxidative polymerization: (1) occurred at $H_2O_2$ overexpressed tumor sites and activated an effective NIR-II PA imaging and PTI; (2) enabled a 6.4-fold higher reaction rate for ADPAH@AOT group than that of free ADPAH group; (3) induced a 3.5-fold temperature increase after ten-minute NIR-II irradiation, offering the potential for dosage mitigation and toxicity reduction. After the assistance of bromelain to digest collage in ECM, the infiltration of T lymphocytes in tumor tissue for ADPAH@AOT-Bro/NIR-II group was enhanced by 2.0-fold for the primary tumor and 2.1-fold for the distant tumor than those of ADPAH@AOT/NIR-II group, respectively. In particular, ADPAH@AOT/NIR-II group also showed the highest antitumor efficiency for both the primary and distant tumors among all groups. We envision that this confined polymerization approach would greatly promote the feasibility of conducting multiple polymerization reactions in living systems to confer new functionalities, modulate biological processes, and even fabricate cellular compartments.

## Methods

### Cell lines and animals

Mouse mammary carcinoma cell 4T1 and NIH-3T3 fibroblasts were obtained from the American Type Culture Collection (ATCC). Balb/c female mice (5-6 weeks) were purchased from InVivos Pte. Ltd. (Singapore). All animal experiments were reviewed and approved with the Guidelines for Care and Use of Laboratory Animals of the Institutional Animal Care and Use Committee of Nanyang Technological University (NTU-IACUC) with a protocol number of A19016.

### Photopolymerization of NaSS and NaSS@AOT

NaSS (50 mM, 200 μL) and Irgacure 2959 (20 mM in methanol, 20 μL) were added into deionized water (2.0 mL). AOT (50 mM, in ethanol, 20 μL) was added to the above solution under vigorous stirring to obtain the NaSS@AOT. Then, NaSS and NaSS@AOT were illuminated using a UV lamp (365 nm) and their emission spectrum (excitation at 480 nm) was recorded every minute.

### Photopolymerization of NaSS and NaSS@AOT in living cells

Murine mammary carcinoma cells (4T1) were grown in RPMI 1640 culture medium containing 10% fetal bovine serum and 1% antibiotics (penicillin and streptomycin) at 37 °C in a 5% $CO_2$ incubator. 4T1 cells were then grown in an eight-well plate ($1 \times 10^4$ cells per well) overnight and then added with NaSS or NaSS@AOT (containing 25 mM of NaSS and 1 mM of Irgacure 2959). After incubation for 4 h, cells were washed three times with PBS solution and then illuminated at 365 nm for 5 min. Finally, the nuclei were stained with Hoechst 33342 and the samples were subjected to CLSM observation.

### Preparation of ADPAH hydrochloride

HCl (6 mol) was slowly added to a solution of monomer (6.0 mol, ADPA) in tetrahydrofuran (THF) solution (10 mL). After stirring for 0.5 h, the sediment was filtrated to get ADPAH as a brown solid. Other substrates (OPD, PPD, MPD, and NDA) in the form of amine hydrochloride salt were obtained using the same method.

### Preparation of ADPAH@AOT

ADPAH (100 mM, 20 μL) and HRP (0.6 μM, 20 μL) were added into deionized water (2.0 mL). Then, AOT (200 mM, in ethanol, 15 μL) was added to the above solution under ultrasound conditions to obtain the ADPAH@AOT. Other monomers@AOT (OPD, PPD, MPD, and NDA) were obtained using the same method.

### $H_2O_2$-responsive polymerization

To study the $H_2O_2$-responsive polymerization, the UV–vis/NIR absorption spectra of OPD@AOT, PPD@AOT, MPD@AOT, ADPA@AOT and NDA@AOT before and after 8 h-incubation with 100 μM $H_2O_2$ were measured. For the kinetics study, all of the above reactions were performed by measuring the absorbance of OPD@AOT (430 nm), PPD@AOT (555 nm), MPD@AOT (490 nm), ADPA@AOT (655 nm), NDA@AOT and ADPAH@AOT (1003 nm) at different reaction time points, and the reaction rate of each polymerization was measured from the slope of the absorbance. For the interference experiment, 10% of fresh cell supernatant was added to the above solution. Then their absorbance at different reaction time points was measured and the reaction rate was calculated.

The further investigation of $H_2O_2$-responsive polymerization of ADPAH@AOT, various concentrations of $H_2O_2$ (0, 10, 20, 50, 100, and 200 μM) were incubated with ADPAH@AOT for 8 h with gentle stirring, followed by UV–vis/NIR absorption spectra measurement. To study the specificity response to $H_2O_2$, ADPAH@AOT (50 μg mL$^{-1}$) was incubated with various reactive oxygen species ($^1O_2$, $ClO^-$, $O_2^{\cdot-}$, $\cdot OH$, and $ONOO^-$) (100 μM) for 10 min with gentle stirring, followed by UV–vis/NIR absorption spectra measurement.

## Computational setup

The initial structures of all molecules were generated by performing conformational searches using Grimme's programs xTB 6.3 and CREST 2.10.2, and further optimized using the (U)B3LYP functional with Grimme's D3(BJ) dispersion correction[28,29]. The 6-31 + G(d) basis sets were chosen to describe all atoms. For open-shell singlet, the keyword 'guess=mix' was used. Frequency calculations were carried out at the same level of theory to identify all of the stationary points as minima (zero imaginary frequency) and to provide the thermal correction to free energies at 298.15 K and 1 atm. For comparing the free energies of different conformations and calculating redox potentials, the higher accuracy single-point energy calculations were performed at the M06-2X level with the 6-311 + G(d,p) for all atoms, using the Solvation Model based on Density (SMD) with water as the solvent. The redox potentials were corrected to $E^0(SHE) = 4.28$ V. The energies of molecular orbital and excitation energy (using TD-DFT) were calculated at the CAM-B3LYP-D3(BJ)/6-311 + G(d,p) level with SMD. All calculations were performed with the Gaussian 16 (Revision A.03) software package.

## Photothermal effect

For the $H_2O_2$-dependent photothermal property, ADPAH@AOT aqueous solution (50 µg mL$^{-1}$) with different concentrations of $H_2O_2$ (0, 10, 20, 50, 100, and 200 µM) was irradiated using a 1064 nm laser (1 W cm$^{-2}$) for 10 min. The temperature of ADPAH@AOT solution was recorded by a thermal infrared imaging camera (FLIR C2, USA). For the concentration-dependent photothermal property, ADPAH@AOT aqueous solution with different concentrations of ADPAH (0, 10, 20, 30, 40, and 50 µg mL$^{-1}$, with $H_2O_2$ 100 µM) was irradiated by 1064 nm laser for 10 min, and the temperature change was recorded by using a thermal camera. Subsequently, ADPAH@AOT solution (50 µg mL$^{-1}$, with the addition of 100 µM $H_2O_2$) with different power densities (0.3, 0.5, 0.75, and 1 W cm$^{-2}$) was irradiated by 1064 nm laser for 10 min, and the temperature was recorded every 30 s. To investigate the photostability of polymerized ADPAH@AOT, 1.0 mL of ADPAH@AOT (50 µg mL$^{-1}$ with the addition of 100 µM) in water was exposed to 1064 nm laser (1 W cm$^{-2}$) irradiation for 10 min, and then cooled down for 15 min. This procedure was performed for four consecutive cycles.

## In vitro cytotoxicity assays

The biocompatibility of ADPAH@AOT was evaluated using MTT (3-(4,5-dimethylthiazol-2-yl)-2,5-diphenyl-2H-tetrazolium bromide) assay. NIH-3T3 fibroblasts were grown onto a 96-well plate (7000 cells/well) and followed by incubation overnight. Then, the medium was replaced with a fresh medium containing ADPAH@AOT of known concentrations. After 24 h, the cells were incubated in a cell medium (100 µL) containing an MTT solution (10 µL). After another 4 h incubation, DMSO (150 µL) was injected to replace the medium. The absorbance was measured by a microplate reader at the 490 nm wavelength. The relative cell viability (%) was determined by comparing the absorbance values of sample wells with that of control wells.

The cytotoxicity of ADPAH and ADPAH@AOT was also evaluated using an MTT assay. 4T1 cells were grown onto a 96-well plate (7000 cells/well) and followed by incubation overnight. Then the medium was replaced with a fresh medium containing ADPAH@AOT of known concentrations (containing $H_2O_2$ 100 µM to mimic the tumor microenvironment). After 24 h, the cells were incubated in a 100 µL cell medium containing MTT solution (10 µL). After another 4 h incubation, DMSO (150 µL) was injected to replace the medium. The absorbance was measured by a microplate reader at the 490 nm wavelength. The relative cell viability (%) was determined by comparing the absorbance values of sample wells with that of control wells. As for the ADPAH@AOT/NIR-II group, 4T1 cells were treated with ADPAH@AOT (20 µg mL$^{-1}$) for 4 h and then exposed to 1064 nm (1 W cm$^{-2}$) light irradiation. After another 20 h incubation, cell viability was detected using an MTT assay kit.

## Immunogenic cell death

For immunofluorescence staining of CRT (or HMGB-1) expression, 4T1 cells were grown onto an eight-well plate (10,000 cells/well) and followed by incubation overnight. Thereafter, 4T1 cells were treated by ADPAH@AOT (20 µg mL$^{-1}$) for 4 h and then exposed to 1064 nm (1 W cm$^{-2}$) laser irradiation. After another 20 h incubation, the supernatants were collected for ATP quantification and the cells were washed, fixed (4% paraformaldehyde for 20 min), permeabilized (0.1% Triton X-100 for 10 min), and blocked (3% BSA for 60 min). Then anti-CRT antibody (or HMGB-1 antibody) was incubated with the cells at 4 °C overnight. After washing with PBS three times, the cells were incubated with the corresponding secondary antibody at room temperature for 2 h. Subsequently, the nuclei were stained with Hoechst 33342 and the samples were observed by CLSM.

## In vitro DC maturation and cytokine analysis

Here, a 24-well transwell system was utilized[34]. Briefly, 4T1 cells in the upper chambers were pre-treated with ADPAH@AOT (40 µg mL$^{-1}$) in the presence of $H_2O_2$ (100 µM) and followed by a NIR-II laser irradiation, then the BMDCs were seeded in the lower chambers. After coincubation for 6 h, the DCs were harvested and co-stained with APC-conjugated anti-mouse CD11c, FITC-conjugated anti-mouse CD80, and PE-conjugated anti-mouse CD86 antibodies, followed by flow cytometry. At the same time, the media supernatants were also collected for cytokines detection.

## PA imaging of polymerization of ADPAH@AOT in vitro and in vivo

ADPAH@AOT solution (50 µg mL$^{-1}$) with different concentrations of $H_2O_2$ (0, 10, 20, 50, 100, 150, and 200 µM) was added into photoacoustic glass tubes to collect the PA intensity under 1064 nm laser. For in vivo PA imaging, ADPAH@AOT solution was intratumorally administrated into 4T1 tumor-bearing mice at the dosage of 3.0 mg kg$^{-1}$ for PA imaging scans under 1064 nm laser irradiation.

## Preparation of and characterization of ADPAH@AOT-Bro

ADPAH (100 mM, 20 µL), HRP (0.6 µM, 20 µL), and bromelain (500 µg mL$^{-1}$, 200 µL) were added into deionized water (2.0 mL). Then, AOT (200 mM, in ethanol, 15 µL) was added to the above solution under vigorous stirring to obtain the ADPAH@AOT-Bro.

The enzymatic activity of ADPAH@AOT-Bro towards gelatin was defined as Gelatin Digestion Unit (GDU): one unit digests 1.0 mg of amino nitrogen from gelatin in 40 min[35]. Typically, gelatin solution (2.5 mL, 1.0% in PBS buffer) was placed into 5 mL tubes. Then, 1.5 mL of ADPAH@AOT or ADPAH@AOT-Bro solution (2 mg mL$^{-1}$) was added into the tubes and the solution was mixed homogeneously. The tubes were exposed to a 1064 nm laser (10 min) and hydrogen peroxide solution (40.0 µL, 3%) was added to stop the reaction. Subsequently, the pH of the solution was adjusted to 6.0 with NaOH and then formaldehyde (0.2 mL, 37%) was added to the solution. Titration was performed with 0.05 N NaOH until the pH reached 7.8 and the titration volume of the test solution was recorded. For the blank, gelatin solution (1.0%, 2.5 mL) was placed into 5 mL tubes containing $H_2O$ (1.5 mL) and then treated with hydrogen peroxide solution (40.0 µL, 3%). Titration was performed as above, and the titration volume of the blank solution was also recorded. Enzymatic activity was calculated as follows: GDU = (test volume−blank volume) × 700/mg enzyme.

## In vivo immune analysis

Mice were subcutaneously injected with 4T1 cells ($1 \times 10^6$ cells) on the right side. After five days, the left flanks were injected with 4T1 cells ($1 \times 10^6$ cells) and designed as distant tumors. Once the primary tumor volume reached ~50 mm$^3$, ADPAH@AOT-Bro was intratumorally injected into the primary tumors at the dosage of 3.0 mg kg$^{-1}$ ($n = 3$). At 4 h post-injection, primary tumors received 1064 nm NIR light

(1 W cm$^{-2}$, 5 min) irradiation. On day 7, mice were sacrificed and blood, tumors, and spleens of mice were harvested. Blood was placed at room temperature for 2 h and then centrifuged at $300 \times g$ for 5 min to obtain serum for IFN-γ detection. The tumors on the right side were collected for CRT and HMGB-1 staining, and the tumors from both sides were collected for CD4 and CD8 immunofluorescence staining. At the same time, the tumors were collected and cut into small pieces and incubated with the solution (containing 1 mg mL$^{-1}$ collagenase I and IV, and 0.2 mg mL$^{-1}$ DNase I) at 37 °C for 2 h, and then the mixture was filtered through a strainer to get single cell suspension. Thereafter, the cell suspensions were co-stained with Alexa Fluor®700 anti-mouse CD45, FITC-conjugated anti-mouse CD3, PE-conjugated anti-mouse CD4, and APC-conjugated anti-mouse CD8 for 20 min, and followed by flow cytometry analysis. Spleen and tumor-draining lymph nodes were pressed gently and filtered through a strainer to get a single-cell suspension. Then, the cell suspensions were stained with APC anti-mouse CD11c, FITC-conjugated anti-mouse CD80, and PE-conjugated anti-mouse CD86, followed by flow cytometry analysis.

### In vivo antitumor efficiency

Mice were subcutaneously injected with 4T1 cells ($1 \times 10^6$ cells) on the right side. After five days, the left flanks were injected with 4T1 cells ($1 \times 10^6$ cells) and designed as distant tumors. Once the primary tumor volume reached ~50 mm$^3$, mice were divided into five groups ($n = 5$ in each group): (1) PBS, (2) ADPAH@AOT, (3) ADPA-H@AOT-Bro, (4) ADPAH@AOT/NIR-II, and (5) ADPAH@AOT-Bro/NIR-II. The primary tumors were intratumorally injected with ADPAH@AOT and ADPAH@AOT-Bro at the dosage of 3.0 mg kg$^{-1}$. At 4 h post-injection, mice in groups (4) and (5) were exposed to a 1064 nm (1 W cm$^{-2}$) light for 5-min irradiation and the temperatures were recorded every minute via a thermal camera. The tumor volume was measured with a caliper every other day, which was calculated as followings: tumor volume (V) = a × b$^2$ / 2, where a is the major axis and b is the minor axis of the tumor. The tumors for both sides were harvested for caspase-3 and H&E staining on day 14 after the mice were sacrificed.

### Statistical analysis

Results were expressed as mean ± SD. Shapiro-wilk test was used to assess the normality of the data. Statistical comparisons were conducted by one-way ANOVA with a Tukey post-hoc test. For all results, *$p < 0.05$, **$p < 0.01$, and ***$p < 0.001$ were considered to be statistically significant. All statistical calculations were conducted using GraphPad Prism V 9.5.0.

### Reporting summary

Further information on research design is available in the Nature Portfolio Reporting Summary linked to this article.

## Data availability

Source data are provided with this paper. The authors declare that the remaining data supporting the findings of this study are available within the article, its Supplementary Information and Source Data file, and the full image dataset is available from the corresponding author upon request. Source data are provided with this paper.

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

## Acknowledgements

This work was supported by the National Natural Science Foundation of China (21972047, W.Z.), the Guangdong Provincial Pearl River Talents Program (2019QN01Y314, W.Z.), the Program for Guangdong Introducing Innovative and Entrepreneurial Teams (2019ZT08Y318, W.Z.), the Natural Science Foundation of Guangdong Province China (2021A1515220051, W.Z.), the Guangdong International Technology Cooperation Project (2022A0505050008, W.Z.), the China Postdoctoral Science Foundation (2022T150221, Yun Chen), the Guangdong Basic and Applied Basic Research Foundation (2022A1515110374, Yun Chen), the National Research Foundation Singapore under Its Competitive Research Programme (NRF-CRP26-2021-0002, Y.Z.), and the Ministry of Education Singapore under the Research Centres of Excellence Scheme (Institute for Digital Molecular Analytics and Science, Y.Z.). Computational work was supported by Center for Computational Science and Engineering at Southern University of Science and Technology.

## Author contributions

Yun Chen, W.Z., and Y.Z. conceived the idea and designed the research. Yun Chen, M.Z., X.C., X.Z., W.Y., and Y.W. conducted the experiments. Yu Chen and P.Y. performed the theoretical arithmetic. Y.Z., W.Z., P.Y., and Yun Chen drafted the manuscript. All authors discussed the results and commented on the manuscript.

## Competing interests

The authors declare no competing interests.
