## [Peer Review File · Nature Communications]

REVIEWER COMMENTS

Reviewer #1 (Remarks to the Author):

In the manuscript entitled “Nanocompartment-Confined Polymerization in Living Systems”, the authors purport the design, characterization, and efficacy validation of a novel strategy for polymerization in the living system capable of cell imaging and photoimmunotherapy. This approach allows for having locally high concentrations of monomers, which can increase the polymerization rate and reduce potential toxicity. Compartmentalization also creates a secluded environment that minimizes the influence of active substances on the polymerization process. As a result, this strategy would greatly increase the probability of successful polymerization in living system. The endogenous oxidative polymerization was capable of activating effective NIR-II photothermal immunotherapy, providing an exciting application for in vivo polymerization. Overall, this study is both novel and rigorous, featuring characteristic data and sufficient discussions. Thus, I recommend its publication following minor revisions.

1. Please provide the power density of UV light used in the photoinduced polymerization.
2. Authors screened out the well-performed compound from different aniline compounds. Why did authors use p-aminodiphenylamine (ADPA) in form of a hydrochloride salt rather than other aniline analogs? What's the function of hydrochloride salts?
3. The biocompatibility of ADPAH@AOT towards normal cells is suggested to be studied. Moreover, will the polymerization of ADPAH@AOT occur in normal cells?
4. Authors are suggested to provide a schematic diagram for a clear understanding of photoimmunotherapy after polymerization.
5. If applicable, please add scale bars to Figure 5a.
6. The statistical analysis of Figure 6b,c seems missing, please specify whether this difference is statistically significant.
7. Please check the format of the references in the manuscript. For example, the page numbers in reference 8 is incomplete, and the title capitalization in references 6 and 22 is different from others.

Reviewer #2 (Remarks to the Author):

In this manuscript, the authors reported a nanocompartment-confined strategy for oxidative polymerization in living systems, where a 6.8- fold higher reaction rate than that of free monomers was observed and an effective second near infrared photoacoustic imaging-guided photothermal

immunotherapy at tumor sites was achieved. There remain a few flaws to be corrected before the acceptance. Here are our suggestions.

Q1 As for the living system the authors mentioned, is it an intracellular microenvironment or an extracellular microenvironment or both? This should be further clarified in the manuscript.

Q2 In the introduction, the passage of 'Despite these encouraging findings, artificial biological polymerization still encounters certain problems: 1) transportation of high concentrations of monomers is always required; 2) the ubiquitous bioactive and metabolic substances in the living system might quench the polymerization process; 3) most polymerizations demand extra stimulates (e.g., ultraviolet (UV) irradiation, metal ions, etc.) to initiate the reaction; and 4) the synthetic polymers suffer from functional limitations to modulate additional biological responses' are quite similar to the passage of 'Progress in this field is mainly limited by the following obstacles: (a) It is difficult to transport concentrated monomers across cell membranes. (b) The polymerization process can be easily quenched by intracellular biological and metabolic substances. For example, dissolved oxygen can quench radical polymerizations, and abundant amino acids can cause side reactions in polycondensation. (c) Most polymerizations require additional irradiation or heat as a trigger, which limits further in vivo applications because the propagation of light or thermal energy through tissues is inefficient' in J. Am. Chem. Soc. 2021, 143, 28, 10709–10717. The related citation should be added.

Q3 Molecular weight is an important factor determining the physical and chemical properties of polymers. Therefore, during the photopolymerization of NaSS and NaSS@AOT, molecular weights of the products are suggested to be characterized by gel permeation chromatography (GPC).

Q4 Raw scale bars can be seen in the bottom right corner of all the images in Figure 4f, which should be removed before adding the new ones.

Q5 Statistical analyses are suggested to be conducted in the tumor growth curves.

Reviewer #3 (Remarks to the Author):

The develop of micelles to encapsulate monomers and initiator , which then enter cells and can be activated photochemically to generate polymers is nice and clearly demonstrated. That said I cannot

really see the green fluorescence in figure 2. As a general comment, the authors need to be more selective as there is often far too much in every figure. They need to think what is important to help the reader and tell the story? Figure 4 is another example – far, far too much data – they need to be far more selective.

Comments such as “addition of fresh cell supernatant, acting as the influencing factor, reduced the rate of polymerization by 54.0% for OPD, 66.7% for PPD, 39.4% for MPD, 91.6% for ADPA, 24.4% for NDA, and 94.3% for ADPAH, respectively. In contrast, only 48.5%, 43.1%, 17.8%, 87.1%, 1.7%, and 78.2% reductions in polymerization rate were observed in nanocompartment-directed groups, respectively” are confused and the errors here will be significant. The decimal places are clearly questionable and is there a real difference between 54% and 49% for OPD?

The authors MUST read the paper - (RSC Adv., 2022, 12, 13154-13167) as in this paper the authors report anionic vesicles made from AOT (as here) and show them to be particularly suitable for HRPC-catalyzed reactions (including with aniline – that did not work here). The authors in this paper also observed emeraldine salt form of linear polyaniline and NIR-II bands (as here). Thus the whole section on endogenous oxidative polymerization needs to be rewritten – and comments like “we were surprised to learn” needs to be replaced with the literature context of others who made these materials previously in these vesicles and have already observed this - this applies to figure 3 and the text around it too. All in all – the previously published work is significant – the novelty here is the in cell work (which is nice) – but the authors must bring in the work of others that reports these previous findings.

Elsewhere comments like “interesting NIR-II absorption” should ref the original paper (RSC Advances) etc..

The light mediated killing is nice and clear – However the authors then go on to say the generation of DAMPs (30.4%), the photoirradiated ADPAH@AOT significantly increased the maturation of DCs to 45.0% - firstly again I have no confidence in such decimal places - and I would want to see many more independent repeats of these experiments before the difference of 30% and 45% is given such prominence.

Comments about Bromelain being a temperature-sensitive protease are a little mis-guided/mis-understood. Reactions are temperature sensitive (simple Arrhenius thermodynamics) – and the same is true for enzymes – they are faster when warmed (as long as they do not denature).

Comments about Bromelain anti-inflammatory, anticancer, and immunomodulatory activities are mis-guided – it is after all a protease, typically from pineapple - comments about supposed anti-inflammatory, anticancer, and immunomodulatory activity from sparse papers add little value.

How does the protease work if it is inside the micelle (ADPAH@AOT-Bro)? How would the enzyme access the collagen? If it leaves the particle, it will not be heated – if it stays how will collagen access the enzyme? There needs to be a significant amount of work done to explain why a protease entrapped in this system would be expected to work/function. This is important as this then leads onto the rest of the paper. Thus, experiments on isolated particles (ADPAH@AOT-Bro) and analysis of labeled protein digestion etc... is needed to show how they work.

Skin is also not particularly optically transparent even at 1060 nm (see the paper by Effect of wavelength and beam width on penetration in light-tissue interaction using computational methods) – the authors should modify their text to discuss this in better detail/accuracy.

The animal data needs a lot more details of things like numbers and powers and a statistician should also be consulted to ensure that the differences (with the small animal numbers used) are statistically significant (in line with current research practice – and journal guidelines). This is beyond the p numbers reported.

I would urge the authors to think about the readers. The paper is so full of abbreviations that it is exceptionally difficult to read and understand in many places. As an example: NIR-II PTT is a promising immune therapy to activate immunogenic cell death (ICD) by the exposure of damage-associated molecular patterns (DAMPs) and promoting dendritic cell (DC) maturation^{21,22}. To evaluate the activation of ICD after irradiation of the synthetic polymer, 4T1 cells were pre-treated with H₂O₂ (100 μM) and followed by treatment of ADPAH@AOT (Fig. 4e). An upregulated calreticulin (CRT) expression was shown in the CLSM images of ADPAH@AOT/NIR-II group, while negligible CRT signals were observed in the control and ADPAH@AOT group (Fig. 4f). Also, the signals of high mobility group box 1 (HMGB-1) were located predominantly in the nuclei in ADPAH@AOT and control group, whereas significantly HMGB-1 was released in the ADPAH@AOT/NIR-II group But much of the manuscript is like this.

Comments like “These issues cause the slow development of polymerization in living systems, which lags far behind the current development of polymerization” add little value – of course polymerization in living systems lags behind current polymerization (the later has had decades of research to develop).

The authors say “Theoretically, any reported polymerization...” – not sure this is correct – anionic for example or condensation ?

Fig. 1 Schematic illustration.... Is it a figure or a scheme or an illustration ? Proteins are not branched structures as shown

photothermal immunotherapy – this suggests an antibody or immune component is used – this is not correct – so it is not immunotherapy.

Why is sodium bis(2-ethylhexyl) sulfosuccinate given the abbreviation AOT ?

I am unsure of the sentence “the analogues but ANI in the presence of H₂O₂”

Comments like “heat generated by photothermal agents” - what is a photothermal agent? They (like any dye) absorb heat and get hot and generate sound – this is basic photoacoustics.

Response to Reviewers' Comments for
Nanocompartment-Confined Polymerization in Living Systems

Reviewer #1:

In the manuscript entitled “Nanocompartment-Confined Polymerization in Living Systems”, the authors purport the design, characterization, and efficacy validation of a novel strategy for polymerization in the living system capable of cell imaging and photoimmunotherapy. This approach allows for having locally high concentrations of monomers, which can increase the polymerization rate and reduce potential toxicity. Compartmentalization also creates a secluded environment that minimizes the influence of active substances on the polymerization process. As a result, this strategy would greatly increase the probability of successful polymerization in living system. The endogenous oxidative polymerization was capable of activating effective NIR-II photothermal immunotherapy, providing an exciting application for in vivo polymerization. Overall, this study is both novel and rigorous, featuring characteristic data and sufficient discussions. Thus, I recommend its publication following minor revisions.

Response: We appreciate Reviewer 1’s positive comments. We have revised the manuscript according to the points raised.

1. Please provide the power density of UV light used in the photoinduced polymerization.

Response: Thanks for the reviewer’s suggestion. The power density of UV light is 2.5 mW cm⁻², which has been added on Page 6 in the revised manuscript.

2. Authors screened out the well-performed compound from different aniline compounds. Why did authors use p-aminodiphenylamine (ADPA) in form of a hydrochloride salt rather than other aniline analogs? What’s the function of hydrochloride salts?

Response: Thanks. There are two reasons that we chose ADPA in the form of a hydrochloride salt (ADPAH): 1) Transforming ADPA to ADPAH can increase its hydrophilicity, thereby improving its ability to be effectively entrapped into the hydrophilic inner compartment; 2) The hydrochloride salt form of ADPAH provides more positive charges, facilitating electrostatic interactions with AOT vesicles, thereby contributing to the absorption in the NIR-II range.

3. The biocompatibility of ADPAH@AOT towards normal cells is suggested to be studied. Moreover, will the polymerization of ADPAH@AOT occur in normal cells?

Response: Thanks for the reviewer’s valuable comment. The biocompatibility of ADPAH@AOT towards NIH-3T3 fibroblasts have been added to the revised manuscript. It can be seen that ADPAH@AOT displayed negligible cytotoxicity against NIH-3T3 cells with a concentration up to 40 µg mL⁻¹, indicating the high compatibility of ADPAH@AOT towards

normal cells (**Fig. 4b**). After 1064 nm laser irradiation, the cell viability of ADPAH@AOT was still greater than 90% even with a concentration of 40 $\mu\text{g mL}^{-1}$, indicating almost no polymerization occurred (**Supplementary Fig. 14**). This phenomenon may be caused by the variance in H_2O_2 concentration levels between normal cells and tumor cells. Especially, the H_2O_2 concentration in tumors can reach up to 100 μM , which is about 100 times higher than the concentration in healthy tissues (*Nat. Rev. Bioeng.* **2023**, *1*, 125-138). (Please check Page 12 for the discussion, and **Supplementary Fig. 14** in the Supplementary Information)

Fig. 4b Relative cell viabilities of NIH-3T3 cells after treatment with ADPAH@AOT at different concentrations (n = 5).

Supplementary Figure 14. Relative cell viabilities of NIH-3T3 cells after treatment with ADPAH@AOT with or without 1064 nm irradiation for 5 min (1 W cm^{-2}) (n = 5).

4. Authors are suggested to provide a schematic diagram for a clear understanding of photoimmunotherapy after polymerization.

Response: We thank the reviewer's suggestion. As shown in **Supplementary Fig. 27**, the H_2O_2 -responsive polymerization activated effective NIR-II PTT at tumor sites, which could

also activate immunogenic cell death induction, promote DC maturation, and assist with ECM degradation bromelain to facilitate the T cell infiltration. The schematic diagram has been added in **Supplementary Fig. 27** and the discussion has been supplemented on Page 17 in the revised manuscript.

Supplementary Figure 27. Proposed pathway of H₂O₂ mediated polymerization of ADPAH@AOT-Bro in living system and their role in NIR-II photothermal immunotherapy.

5. If applicable, please add scale bars to Figure 5a.

Response: We thank the reviewer's suggestion. The scale bar has been added in **Fig. 5a** and the corresponding changes are indicated in the revised figure below.

Fig. 5a *In vivo* PA imaging of the 4T1-bearing tumor mice after intertumoral administration of ADPAH@AOT-Bro (1064 nm, n = 3).

6. The statistical analysis of Figure 6 b,c seems missing, please specify whether this difference is statistically significant.

Response: Thanks for the reviewer's suggestion. The statistical analyses have been conducted in the tumor growth curves and the corresponding changes are indicated in the

revised figure below.

Fig. 6b,c Tumor growth curves of **b** primary and **c** distant tumors in 4T1-tumor-bearing mice following various treatments (n = 5). ns: not significant, ** $p < 0.01$, and *** $p < 0.001$.

7. Please check the format of the references in the manuscript. For example, the page numbers in reference 8 is incomplete, and the title capitalization in references 6 and 22 is different from others.

Response: Thanks for the reminder. We have checked all the references and revised the references 6, 8, and 22.

Reviewer #2:

In this manuscript, the authors reported a nanocompartment-confined strategy for oxidative polymerization in living systems, where a 6.8- fold higher reaction rate than that of free monomers was observed and an effective second near infrared photoacoustic imaging-guided photothermal immunotherapy at tumor sites was achieved. There remain a few flaws to be corrected before the acceptance. Here are our suggestions.

Response: We thank the reviewer for the careful consideration of our manuscript and for providing very useful comments, suggestions, and recommendations. We have revised the manuscript according to the points raised.

1. As for the living system the authors mentioned, is it an intracellular microenvironment or an extracellular microenvironment or both? This should be further clarified in the manuscript.

Response: Thanks for the reviewer's suggestion. The living system refers to both the intracellular and extracellular microenvironment. For instance, H₂O₂ acts as both extracellular and intracellular signaling molecule (*Nat. Rev. Bioeng.* **2023**, *1*, 125-138; *Nano-Micro Lett.* **2020**, *12*, 15). The detailed descriptions about living system have been added to the revised manuscript (Please check Page 3 for the discussion).

2. In the introduction, the passage of 'Despite these encouraging findings, artificial biological polymerization still encounters certain problems: 1) transportation of high concentrations of monomers is always required; 2) the ubiquitous bioactive and metabolic substances in the living system might quench the polymerization process; 3) most polymerizations demand extra stimulates (e.g., ultraviolet (UV) irradiation, metal ions, etc.) to initiate the reaction; and 4) the synthetic polymers suffer from functional limitations to modulate additional biological responses' are quite similar to the passage of 'Progress in this field is mainly limited by the following obstacles: (a) It is difficult to transport concentrated monomers across cell membranes. (b) The polymerization process can be easily quenched by intracellular biological and metabolic substances. For example, dissolved oxygen can quench radical polymerizations, and abundant amino acids can cause side reactions in polycondensation. (c) Most polymerizations require additional irradiation or heat as a trigger, which limits further in vivo applications because the propagation of light or thermal energy through tissues is inefficient' in *J. Am. Chem. Soc.* **2021**, *143*, *28*, 10709–10717. The related citation should be added.

Response: Thanks for the reviewer's suggestion. The reference has been cited as Ref. 5 in the revised manuscript.

3. Molecular weight is an important factor determining the physical and chemical properties of polymers. Therefore, during the photopolymerization of NaSS and NaSS@AOT, molecular

weights of the products are suggested to be characterized by gel permeation chromatography (GPC).

Response: Thanks for the reviewer's suggestion. The photopolymerization of NaSS generated poly(NaSS) with a number average molecular weight (M_n) of 5437 g mol⁻¹ and a diversity (\mathcal{D}), defined as the ratio of weight-average molecular weight to M_n of 1.82 (**Fig. RL1a**). With the assistant of AOT vesicle, a lower M_n of 5241 g mol⁻¹ and a smaller \mathcal{D} of 1.70 were observed (**Fig. RL1b**).

Fig. RL1. GPC curves of **a** NaSS and **b** NaSS@AOT after photopolymerization.

4. Raw scale bars can be seen in the bottom right corner of all the images in Figure 4f, which should be removed before adding the new ones.

Response: Thanks for the reviewer's suggestion. The raw scale bars have been removed and the corresponding changes are indicated in the revised figure below.

Fig. 4d,e Immunofluorescence imaging of **d** CRT and **e** HMGB-1 in 4T1 tumor cells after treatment with ADPAH@AOT with and without 1064 nm laser irradiation at 1 W cm⁻² (n = 3).

5. Statistical analyses are suggested to be conducted in the tumor growth curves.

Response: Thanks for the reviewer's suggestion. The statistical analyses have been conducted in the tumor growth curves and the corresponding changes are indicated in the

revised figure below.

Fig. 6b,c Tumor growth curves of **b** primary and **c** distant tumors in 4T1-tumor-bearing mice following various treatments (n = 5). *ns*: not significant, ***p* < 0.01, and ****p* < 0.001.

Reviewer #3:

1. The develop of micelles to encapsulate monomers and initiator, which then enter cells and can be activated photochemically to generate polymers is nice and clearly demonstrated. That said I cannot really see the green fluorescence in figure 2. As a general comment, the authors need to be more selective as there is often far too much in every figure. They need to think what is important to help the reader and tell the story? Figure 4 is another example – far, far too much data – they need to far more selective.

Response: We extend our sincere appreciation to the reviewer for bringing forth these invaluable criticisms, suggestions, and recommendations. We have revised the manuscript according to the points raised. We genuinely hope that the reviewer will find our revisions satisfactory.

Thanks for the reviewer's suggestion. To avoid intricate picture typesetting, certain adjustments have been made to the figures to improve clarity and conciseness. For instance, the photothermal and photoacoustic characterizations of Poly-ADPAH from **Fig. 4** have been relocated to **Fig. 3**. As a result, **Fig. 4** exclusively presents data related to endogenous oxidative polymerization in living cells. Additionally, the thermal images of 4T1-tumor-bearing mice and immunofluorescence staining images of collagen I and CRT in **Fig. 5** have been moved to the Supplementary Information section (**Supplementary Figs. 20 and 21**). Notably, the quantitative analysis associated with these images is still kept in **Fig. 4**. Please check the reorganized figures in the revised manuscript.

In **Fig. 2e**, the weak green fluorescence may be not easy to distinguish. We have reconducted this experiment and improved the concentration of NaSS@AOT to 25 mM to get brighter green signals. The corresponding changes are indicated in the revised figure below (Please check Pages 6 and 7).

Fig. 2e,f e CLSM images of 4T1 cells following various treatments (n = 3). Blue fluorescent signal was from Hoechst 33342 cell nucleus staining and green fluorescent signal indicated the formation of poly(NaSS), respectively. **f** Mean fluorescence intensity (MFI) of polymerized NaSS level after various treatments (n = 3). *** $p < 0.001$.

2. Comments such as “addition of fresh cell supernatant, acting as the influencing factor, reduced the rate of polymerization by 54.0% for OPD, 66.7% for PPD, 39.4% for MPD, 91.6%

for ADPA, 24.4% for NDA, and 94.3% for ADPAH, respectively. In contrast, only 48.5%, 43.1%, 17.8%, 87.1%, 1.7%, and 78.2% reductions in polymerization rate were observed in nanocompartment-directed groups, respectively” are confused and the errors here will be significant. The decimal places are clearly questionable and is there a real difference between 54% and 49% for OPD?

Response: Thanks for the reviewer’s suggestion. We understand that, in many cases, when it comes to decimal places, numbers are frequently presented with excessive precision. According to the guidelines established by Cole for presenting numerical data, there are specific recommendations to follow when dealing with percentages (*Arch. Dis. Child.* **2015**, *100*, 608-609; *F1000Research* **2018**, *7*, 450).

These guidelines are as follows:

- 1) Integers or one decimal place for values under 10%, e.g., 1.1%;
- 2) Integers for values above 10%, e.g., 22% not 22.2%;
- 3) One decimal place may be needed for values between 90% to 100% when 100% is a natural upper bound, for example the sensitivity of a test, e.g., 99.9% not 100%;
- 4) Use two or more decimal places only if the range of percents being compared is less than 0.1%, e.g., 50.50% versus 50.55%.

When comparing group means or percentages in tables, rounding should not blur the differences between them. This is the basis for the Hopkins two digits rule, whereby the mean has enough decimal places to ensure two significant digits for the standard deviation (SD) (*Scand. J. Med. Sci. Sports* **2011**, *21*, 867-868). Following these guidelines, we have carefully reviewed the data presented in the manuscript and made necessary revisions to ensure they meet the recommended standards.

To assess the statistical significance of the data, we have conducted the experiments for at least independent three repeats and calculated the errors. As depicted in **Fig. 3d**, the introduction of fresh cell supernatant, which acted as an influencing factor, had a more significant impact on monomers that lacked vesicles. Specifically, after interference, ADPAH@AOT retained $27 \pm 7.1\%$ of its original activity, while the nanocompartment-deficient ADPAH group only reserved $4.8 \pm 0.3\%$ of its activity. In addition, following three separate repetitions, the initial velocity ratio of OPD to iOPD (OPD with interference) was determined to be $49 \pm 1.4\%$. Similarly, the initial velocity ratio of OPD@AOT to iOPD@AOT (OPD@AOT with interference) was calculated to be $59 \pm 4.6\%$. As shown in **Fig. RL2**, a two-tailed Student's *t*-test was performed to assess the statistical significance. The resulting *p*-value was calculated as 0.019, which indicates statistical significance ($p < 0.05$). The above discussions have been added to the revised manuscript (please check pages 8 and 9 for the discussion).

Fig. RL2 Percentage activity of V_{IOPD}/V_{OPD} and $V_{IOPD@AOT}/V_{OPD@AOT}$. Statistical significance was calculated via a two-tailed Student's *t*-test. * $p < 0.05$.

3. The authors MUST read the paper - (RSC Adv., 2022, 12, 13154-13167) as in this paper the authors report anionic vesicles made from AOT (as here) and show them to be particularly suitable for HRPC-catalyzed reactions (including with aniline – that did not work here). The authors in this paper also also observed emeraldine salt form of linear polyaniline and NIR-II bands (as here). Thus the whole section on endogenous oxidative polymerization needs to be rewritten – and comments like “we were surprised to learn” needs to be replaced with the literature context of others who made these materials previously in these vesicles and have already observed this - this applies to figure 3 and the text around it too. All in all – the previously published work is significant – the novelty here is the in cell work (which is nice) – but the authors must bring in the work of others that reports these previous findings.

Response: Thanks so much for the reviewer's suggestion. While there are some studies that have employed similar polymerization to achieve NIR-II absorption *in vitro*, conducting unnatural polymerizations within the biological environment offers unparalleled opportunities for regulating biological processes and imparting novel functionalities. For instance, Deisseroth and Bao et al. genetically engineered specific living neurons to synthesize this similar electrical polyaniline polymers to regulate the electrophysiological behaviours (*Science* **2020**, 367, 1372-1376). Our main objective here is to highlight the significance of employing a nanocompartment-confined strategy for polymerization within a living system. More importantly, the overexpressed H_2O_2 in tumor tissues makes ADPAH@AOT an attractive system for activating NIR-II photoacoustic imaging-guided photothermal immunotherapy at tumor sites.

Our comments like “we were surprised to learn” may inadvertently create the impression that we were the first to report on ADPA polymerization with NIR-II absorption. To avoid the misleading, the literature context of this phenomenon has been added to the revised manuscript. Among these substrates, ADPAH@AOT revealed an obvious NIR-II absorption with a peak at approximately 1003 nm upon exposure to H_2O_2 . This observation serves as an

indication of the presence of polyaniline in its emeraldine salt form (PANI-ES), consistent with findings from other studies (*Science* **2020**, *367*, 1372-1376; *RSC Adv.* **2022**, *12*, 13154-13167; *ACS Catal.* **2014**, *4*, 3421-3434). Briefly, HRP catalyzed H₂O₂-dependent one-electron oxidation with ADPAH to the formation of the diimine, which was unstable in aqueous solutions and underwent a coupling reaction with each other to form Poly-ADPAH (PADPAH) (**Supplementary Fig. 7**). The above discussions and the reference (*RSC Adv.* **2022**, *12*, 13154-13167) have been added to the revised manuscript (please check page 9 and Ref. 18).

4. Elsewhere comments like “interesting NIR-II absorption” should ref the original paper (RSC Advances) etc..

Response: Thanks for the reviewer’s suggestion. This sentence has been revised as follows: “the overexpressed H₂O₂ in tumor tissues makes ADPAH@AOT an attractive candidate for activating NIR-II photothermal therapy (PTT)”. Also, the mentioned paper (*RSC Adv.* **2022**, *12*, 13154-13167) has been referenced in the description of the NIR-II absorption of ADPAH@AOT (Please check page 10 and Ref. 18).

5. The light mediated killing is nice and clear – However the authors then go on to say the generation of DAMPs (30.4%), the photoirradiated ADPAH@AOT significantly increased the maturation of DCs to 45.0% - firstly again I have no confidence in such decimal places - and I would want to see many more independent repeats of these experiments before the difference of 30% and 45% is given such prominence.

Response: Thanks for the reviewer’s suggestion. It should be noted that the average maturation rates of 30% and 45% were obtained from three independent repeats, as stated in **Fig. 4i (Supplementary Fig. 12** in the former version). **Fig. 4h** depicts a representative flow cytometry plot of DC maturation. To make the expression clearly, we used the “mean ± SD” to describe the data. For instance, the above-mentioned sentence has been rephrased as follows: “In contrast to the ADPAH@AOT group, where only a weak maturation rate was observed (30 ± 0.9%), the photoirradiated ADPAH@AOT significantly increased the maturation of DCs to 45 ± 1.4% (**Fig. 4h, i**)” (please check Page 13). In addition, other similar expressions have also been revised accordingly throughout the revised manuscript. The decimal places here also adhere to the guidelines mentioned in Comment #2.

6. Comments about Bromelain being a temperature-sensitive protease are a little misguided/mis-understood. Reactions are temperature sensitive (simple Arrhenius thermodynamics) – and the same is true for enzymes – they are faster when warmed (as long as they do not denature).

Response: Thanks for the reviewer’s suggestion. Typically, each enzyme has an optimal reaction temperature. The optimal temperature of bromelain is around 45 °C, which makes it suitable for a photothermal-mediated enzyme switch in efficient collagen digestion. The

discussion has been added on Page 15.

7. Comments about Bromelain anti-inflammatory, anticancer, and immunomodulatory activities are mis-guided – it is after all a protease, typically from pineapple - comments about supposed anti-inflammatory, anticancer, and immunomodulatory activity from sparse papers add little value.

Response: Thanks for the reviewer's suggestion. Previous *in vitro* and *in vivo* studies have demonstrated that bromelain exhibits anti-inflammatory, anticancer, and immunomodulatory activity (*Front. Oncol.* **2022**, *12*, 1068778; *World J. Pharm. Pharm. Sci.* **2019**, *8*, 488-500; *Biotechnol. Res. Int.* **2012**, *2012*, 976203). In our manuscript, the ADPAH@AOT-Bro group (without NIR-II irradiation) showed increased CRT expression, DC maturation, T cell infiltration, and antitumor activities compared with those of ADPAH@AOT group (**Fig. 5 and Fig. 6**), verifying the biological activity of bromelain. Herein, more discussions about the biological activity of bromelain have been added in the revised manuscript (please check Pages 15-17).

8. How does the protease work if it is inside the micelle (ADPAH@AOT-Bro)? How would the enzyme access the collagen? If it leaves the particle, it will not be heated – if it stays how will collagen access the enzyme? There needs to be a significant amount of work done to explain why a protease entrapped in this system would be expected to work/function. This is important as this then leads onto the rest of the paper. Thus, experiments on isolated particles (ADPAH@AOT-Bro) and analysis of labeled protein digestion etc... is needed to show how they work.

Response: Bromelain shows its function towards collagen after released from AOT vesicle. The electrostatic attractions between anionic interfaces of AOT and in situ formed PADPAH would disturb the balance in the lipid-membrane to cause the release of bromelain (*J Control. Release* **2022**, *352*, 460-471; *Acc. Chem. Res.* **2022**, *55*, 2882-2891). As shown in **Fig. 3e**, TEM image indicated that there were various small black nanoparticles generated near the vesicle membrane after responding to H₂O₂, verifying the interaction of PADPAH with the vesicle templates. Additionally, there was a partial leakage of black dots from the AOT vesicle, indicating a disruption in its nanostructure. After irradiated by an NIR-II light, the resulting PADPAH can harvest the energy from the light and convert it into heat to increase the temperature of the surrounding environment. As shown in the **Supplementary Fig. 17**, the whole tumor could be heated with an increased temperature. Hence, the bromelain in the tumor microenvironment could also be heated to increase its hydrolase activity.

The proteolytic activity of ADPAH@AOT-Bro before and after NIR-II irradiation was detected by using gelatin as the substrate. The collage degradation efficacy of ADPAH@AOT-Bro/NIR-II group, expressed as gelatin digestion units (GDU), was calculated to be 3253.6 GDU g⁻¹, higher than that without NIR-II laser irradiation (1749.6 GDU g⁻¹) (**Supplementary**

Fig. 18). These results confirmed that the enzymatic activity of ADPAH@AOT-Bro could be triggered by the photothermal heating, as the solution temperature increased circa 24.8 °C under NIR-II laser irradiation (**Supplementary Fig. 19**). The above discussions and the figures have been added to the revised manuscript (Please check Pages 15 and 16 for the discussion).

9. Skin is also not particularly optically transparent even at 1060 nm (see the paper by Effect of wavelength and beam width on penetration in light-tissue interaction using computational methods) – the authors should modify their text to discuss this in better detail/accuracy.

Response: Thanks for the reviewer’s suggestion. We will answer this question from the following two aspects: first, 99% of visible light energy would be absorbed by the tissues underneath the skin <2 mm. NIR-II lights, characterized by a wavelength range of 1000 to 1350 nm, exhibits deeper penetration through skin tissues (about 20 mm). This enhanced penetration is attributed to the reduced absorption and scattering effects caused by the skin tissues (*Nat. Commun.* **2022**, *13*, 6596; *Nat. Nanotech.* **2009**, *4*, 710-711; *Lasers Med. Sci.* **2017**, *32*, 1909-1918) (**Fig. RL3a**). Second, the results of photoacoustic imaging (**Fig. RL3b**) and thermal imaging (**Fig. RL3c**) of 4T1-tumor-bearing mice have demonstrated that 1064 nm irradiation can penetrate deep tumor tissue through skin and cause heat effect. So far, there have been a plenty of studies applying 1064 nm laser for photothermal therapy (*Chem. Soc. Rev.* **2021**, *50*, 1111-1137). Therefore, there should be no worries for skin penetration of 1064 nm laser in our experiments. This discussion has been added to the revised manuscript (Please check Page 10).

Fig. RL3 a Absorption coefficient of light with different wavelengths and electromagnetic

radiation (10^2 – 10^9 nm) by the biological constituents within skin tissues. The absorption curve of melanin is obtained by fitting calculations of some known values (*Nat. Commun.* **2022**, *13*, 6596). **b** *In vivo* PA imaging of the 4T1-bearing tumor mice after intertumoral administration of ADPAH@AOT-Bro (1064 nm, n = 3) **c** Thermal images of 4T1-tumor-bearing mice under 1064 nm irradiation (1 W cm^{-2}) at 4 h post intertumoral injection of PBS, ADPAH@AOT, and ADPAH@AOT-Bro (n = 3).

10. The animal data needs a lot more details of things like numbers and powers and a statistician should also be consulted to ensure that the differences (with the small animal numbers used) are statistically significant (in line with current research practice – and journal guidelines). This is beyond the p numbers reported.

Response: Thanks for the reviewer’s suggestion. The detailed description of the animal data has been added in the corresponding figure captions (**Fig. 5** and **Fig. 6**) and the *Methods* section of the main manuscript. We have also consulted a statistician and checked throughout the manuscript accordingly to ensure that the differences are statistically significant.

11. I would urge the authors to think about the readers. The paper is so full of abbreviations that it is exceptionally difficult to read and understand in many places. As an example: NIR-II PTT is a promising immune therapy to activate immunogenic cell death (ICD) by the exposure of damage-associated molecular patterns (DAMPs) and promoting dendritic cell (DC) maturation^{21,22}. To evaluate the activation of ICD after irradiation of the synthetic polymer, 4T1 cells were pre-treated with H_2O_2 (100 μM) and followed by treatment of ADPAH@AOT (Fig. 4e). An upregulated calreticulin (CRT) expression was shown in the CLSM images of ADPAH@AOT/NIR-II group, while negligible CRT signals were observed in the control and ADPAH@AOT group (Fig. 4f). Also, the signals of high mobility group box 1 (HMGB-1) were located predominantly in the nuclei in ADPAH@AOT and control group, whereas significantly HMGB-1 was released in the ADPAH@AOT/NIR-II group But much of the manuscript is like this.

Response: Thanks for the reviewer’s suggestion. These abbreviations are proprietary terms commonly used in the field of biology. We have checked the whole manuscript and revised the related expressions for better readability. For instance, the above-mentioned sentence has been revised as follows: “NIR-II PTT is a promising immune therapy to activate immunogenic cell death by the exposure of damage-associated molecular patterns (DAMPs), e.g., calreticulin (CRT), high mobility group box 1 (HMGB-1), and adenosine triphosphate (ATP). To evaluate the activation of immunogenic cell death after irradiation of the synthetic polymer, 4T1 cells were pre-treated with H_2O_2 (100 μM) and followed by treatment of ADPAH@AOT. An upregulated CRT expression was shown in the CLSM images of ADPAH@AOT/NIR-II group, while negligible red signals were observed in the control groups

(**Fig. 4d**). Also, the signals of HMGB-1 were located predominantly in the nuclei in the control groups, whereas significantly released in the ADPAH@AOT/NIR-II group (**Fig. 4e**). Moreover, the extracellular ATP levels in ADPAH@AOT/NIR-II were 1.6- and 2.2-fold higher than that of ADPAH@AOT and control groups, respectively (**Fig. 4f**)” (Please check Page 12).

12. Comments like “These issues cause the slow development of polymerization in living systems, which lags far behind the current development of polymerization” add little value – of course polymerization in living systems lags behind current polymerization (the later has had decades of research to develop).

Response: Thanks for the reviewer’s suggestion. We have rephrased the sentence as follows: “The existence of these challenges hampers the pace of polymerization in living systems, thereby motivating researchers to actively seek solutions”.

13. The authors say “Theoretically, any reported polymerization...” – not sure this is correct – anionic for example or condensation?

Response: Thanks for the reviewer’s suggestion. Some polymerizations require specific conditions to facilitate the completion of the reaction. For instance, some polymerizations only occur in an anhydrous and oxygen-free environment, which could not be achieved by only nanocompartment-confined strategy. The statement has been revised as follows: “this nanocompartment-confined strategy significantly enhances the feasibility of achieving artificial polymerizations in living systems” (please check Page 4).

14. Fig. 1 Schematic illustration.... Is it a figure or a scheme or an illustration? Proteins are not branched structures as shown

Response: Thanks for the reviewer’s suggestion. In fact, Fig. 1 is a Scheme. *Nature Communications* has a specific formatting requirement: “Please note that schemes are not used; these should be presented as figures”. So, we changed figure caption as follows: “**Fig. 1** Schematic diagram of polymerization in living systems”.

We have also revised the structures of proteins to a conventional form. The corresponding changes are indicated in the revised figure below.

Fig. 1 Schematic diagram of polymerization in living systems. **a** Biological polymerization of monosaccharides, amino acids, and nucleotides to the synthesis of polysaccharides, proteins, and nucleic acids to constitute the elementary components, confer essential functionalities, and modulate the biological process. **b** Nanocompartment-confined strategy to make a broad spectrum of polymerizations achievable in living systems: exogenous photopolymerization of the sodium salt of 4-styrenesulfonate (NaSS) and endogenous oxidative polymerization of p-aminodiphenylamine hydrochloride (ADPAH). NIR-II PA imaging: second near-infrared photoacoustic imaging, PTI: photothermal immunotherapy.

15. photothermal immunotherapy – this suggests an antibody or immune component is used – this is not correct – so it is not immunotherapy.

Response: Photothermal immunotherapy, a novel concept combining photothermal therapy with immunotherapy, can lead to a synergistic thermal-immune effect and trigger a specific antitumor immunity (*Nat. Rev. Mater.* **2019**, *4*, 398-414; *Nat. Rev. Clin. Oncol.* **2020**, *17*, 657-674; *Theranostics* **2021**, *11*, 2218-2231). In recent years, a plenty of studies revealed that photothermal process in an appropriate temperature gradient can noninvasively ablate cancer and efficiently trigger cancer immunogenic cell death to ignite antitumor immunity (*ACS Nano* **2023**, *17*, 8183-8194; *Adv. Mater.* **2020**, 2004788). Until now, a certain number of literatures have achieved photothermal immunotherapy without using any antibody or immune component (*Adv. Mater.* **2021**, *33*, e2008061; *Biomaterials* **2021**, *277*, 121130; *ACS Nano* **2020**, *14*, 2847-2859; *Adv. Funct. Mater.* **2020**, *30*, 1909745; *Small* **2018**, *14*, e1800678;

J. Mater. Chem. B **2019**, *7*, 7406-7414). In our manuscript, we have demonstrated that NIR-II photothermal therapy has the ability to induce immunogenic cell death and facilitate dendritic cell maturation. Therefore, it is appropriate to classify it as photothermal immunotherapy. Thanks for understanding.

16. Why is sodium bis(2-ethylhexyl) sulfosuccinate given the abbreviation AOT?

Response: Thanks for the comments. AOT is an abbreviation commonly used for Aerosol-OT, a trade name that has become the preferred term to refer to sodium bis(2-ethylhexyl) sulfosuccinate (*Chem. Phys. Lipids* **1977**, *18*, 84-104). The detail description has been added in the revised manuscript (please check Page 6).

17. I am unsure of the sentence “the analogues but ANI in the presence of H₂O₂”

Response: Thanks for the comments. HRP was unable to polymerize aniline monomers in our reaction system, a similar phenomenon observed in a previous study (*Science* **2020**, *367*, 1372-1376), which may be ascribed to the higher calculated oxidation potential of ANI (1.16 V). Since ANI did not work in our system, we deleted the description of ANI to avoid the confusion. The sentence has been rewritten as follow: “After the addition of H₂O₂, the polymerization occurred as revealed by the changed ultraviolet/visible/near-infrared absorption spectra (UV-vis/NIR)” (please check Page 8).

18. Comments like “heat generated by photothermal agents” - what is a photothermal agent? They (like any dye) absorb heat and get hot and generate sound – this is basic photoacoustics.

Response: Thanks for the comments. A photothermal agent is a substance or material that can harvest the energy from light and convert the energy into heat to increase the temperature of the surrounding environment. The photothermal effect can also generate acoustic waves that can be detected and converted into imaging signals, which is called photoacoustic (PA) imaging (*Chem. Soc. Rev.* **2019**, *48*, 2053-2108). Hence, the sentence has been rewritten as follow: “As the photothermal effect could generate detectable acoustic waves that can be converted into imaging signals, the possibility of using photoacoustic (PA) imaging for the H₂O₂-triggered polymerization was further examined” (please check page 10).

REVIEWER COMMENTS

Reviewer #1 (Remarks to the Author):

I have carefully looked into the response letter, as well as the updated manuscript and supplementary materials. It seems that the authors have managed to address all of the comments, and the improved version of the manuscript have added data to better support their conclusions. Therefore I am happy to recommend acceptance of the current version for publication.

Reviewer #2 (Remarks to the Author):

I am satisfied with the revisions the authors have made. This manuscript can be accepted at present form.

Reviewer #3 (Remarks to the Author):

I am largely happy with their responses., However their response to light penetration needs to take into account scattering, which is of much more significance than absorption. The graph they give is just absorbance - and most of this is beyond 1500 nm so not really relevant.

I also think that the details from their statistician need to be given to validate the powering and statistical significance for the animal experiments.

The text and the discussion of the micelle (anionic vesicles made from AOT) needs to be more humble with respect to the RSC Advances paper (especially considering the fact this RSC Advances showed that this was suitable for HRPC-catalyzed reactions and observed the emeraldine salt form of linear polyaniline.

Response to Reviewers' Comments

Nanocompartment-Confined Polymerization in Living Systems

Reviewer #1: I have carefully looked into the response letter, as well as the updated manuscript and supplementary materials. It seems that the authors have managed to address all of the comments, and the improved version of the manuscript have added data to better support their conclusions. Therefore I am happy to recommend acceptance of the current version for publication.

Response: Thanks so much for your recommendation of publication.

Reviewer #2: I am satisfied with the revisions the authors have made. This manuscript can be accepted at present form.

Response: Thanks so much for your recommendation of publication.

Reviewer #3:

1. I am largely happy with their responses., However their response to light penetration needs to take into account scattering, which is of much more significance than absorption. The graph they give is just absorbance - and most of this is beyond 1500 nm so not really relevant.

Response: Thanks for the reviewer's suggestion. Generally, the incident light energy is mainly diminished through three processes after penetrating tissues, including reflection, scattering, and absorption by endogenous absorbers (**Fig. RL1a**) (*Small Methods* **2019**, *3*, 1900553). Due to the lower scattering coefficients of various tissues, NIR-II light exhibits enhanced energy preservation and enables greater depth of tissue penetration (**Fig. RL1b**) (*Nat. Biomed. Eng.* **2017**, *1*, 0010). In addition, the main endogenous absorbers including water, blood, fat, and skin show a relatively low absorption in the NIR-II window (**Fig. RL1c**) (*Nat. Nanotechnol.* **2009**, *4*, 710-711). To date, NIR-II light has been widely used for NIR-II photoacoustic imaging, NIR-II fluorescence imaging, and NIR-II photothermal therapy (*Chem. Soc. Rev.* **2021**, *50*, 1111-1137). The weakened light attenuation and the remission of heat generated by endogenous absorbers in the NIR-II window will enhance the signal-to-noise ratio for photoacoustic imaging, minimize side effects for PTT, and improve the penetration depth for both (**Fig. RL1d**). Moreover, biological tissues often show higher tolerance for NIR-II light, thus allowing higher maximum permissible exposure for skin relative to that for NIR-I light. Therefore, in the NIR-II window, the combination of low absorption by main endogenous absorbers and the reduced scattering from various tissues facilitates numerous advantages, including deep tissue penetration capabilities (at least 20 mm), high maximum permission energy, and excellent sensitivity. This discussion has been added to the revised manuscript (Please check Page 10).

Fig. RL1 Comparison between NIR-II versus NIR-I window for the light performance. **a** Schematic illustration of light-tissue interaction. **b** Scattering coefficients (μ 's) of various tissues plotted as a function of wavelength ranging from 400 to 1700 nm. **c** Effective attenuation coefficients of blood, skin, and fat plotted as a function of wavelength ranging from 200 to 1800 nm. **d** Schematic illustration of the deeper penetration depth and higher MPE of the NIR-II light over the light with shorter wavelengths (*Small Methods* **2019**, *3*, 1900553).

2. I also think that the details from their statistician need to be given to validate the powering and statistical significance for the animal experiments.

Response: Thanks for the reviewer's suggestion. The details about the statistical comparisons are summarized as follow: First, Shapiro-wilk test is used to assess the normality of the data (sample size between 3 and 5000, *J. R. Stat. Soc. Ser. C Appl. Stat.* **1982**, *31*, 115-124; *J. Statist. Model. Anal.* **2011**, *2*, 21-33). If the p -value associated with the test statistic is greater than 0.05, it suggests that the data follow a normal distribution and are suitable for one-way ANOVA (otherwise nonparametric test). Then, the F-test is used to determine if the variances are homogeneous. If the variances are homogeneous, one-way ANOVA is performed for comparing means among multiple groups. Otherwise, the Brown-Forsythe and Welch ANOVA are used. For all results, * $p < 0.05$, ** $p < 0.01$, and *** $p < 0.001$ were considered to be statistically significant. All statistical calculations were conducted using GraphPad Prism V. 9.5.0. The details of the p value in each figure have been added in the corresponding figure captions. Take **Fig. 6** as an example, the corresponding changes are indicated in the revised figure caption below.

Fig. 6 *In vivo* antitumor performance of ADPAH@AOT-Bro. **a** Schematic illustration of *in vivo* antitumor schedule. **b, c** Tumor growth curves of **b** primary and **c** distant tumors in 4T1-tumor-bearing mice following various treatments (1064 nm, 5 min, n = 5). ADPAH-Bro@AOT/NIR-II versus ADPAH@AOT/NIR-II in **b**: $p = 0.373$; ADPAH-Bro@AOT/NIR-II versus other three groups in **b**: $p < 0.001$; ADPAH-Bro@AOT/NIR-II versus ADPAH@AOT/NIR-II in **c**: $p = 0.002$. ADPAH-Bro@AOT/NIR-II versus other three groups in **c**: $p < 0.001$. **d** Immunofluorescent images of caspase-3 (red) in primary and distant tumor sections following various treatments. Quantification of expression level of caspase-3 in **e** primary and **f** distant tumors sections from euthanized mice following various treatments (n = 3). ADPAH-Bro@AOT/NIR-II versus ADPAH@AOT/NIR-II in **e**: $p = 0.005$. ADPAH-Bro@AOT/NIR-II versus other three groups in **e**: $p < 0.001$; ADPAH-Bro@AOT/NIR-II versus other groups in **f**: $p < 0.001$. **g** H&E-stained images of the primary and abscopal tumor sections from euthanized mice following various treatments (n = 3). Data were presented as mean \pm SD. Statistical significance in **b, c, e** and **f** was calculated via one-way ANOVA followed by Tukey post-hoc test. ns: not significant, ** $p < 0.01$, and *** $p < 0.001$.

3. The text and the discussion of the micelle (anionic vesicles made from AOT) needs to be more humble with respect to the RSC Advances paper (especially considering the fact this RSC Advances showed that this was suitable for HRPC-catalyzed reactions and observed the emeraldine salt form of linear polyaniline.

Response: Thanks for the reviewer's suggestion. After introduction of HRP, the description of the similar reaction was added as follows: Previous research has demonstrated the effectiveness of anionic vesicles derived from AOT in facilitating HRP-catalyzed reactions, particularly when utilizing aniline analogues as the monomer (Please check Page 8);

Moreover, regarding the NIR-II absorption after polymerization, the pioneer work of RSC Advance paper was cited as follow: Among these substrates, ADPAH@AOT revealed an obvious NIR-II absorption with a peak at approximately 1003 nm upon exposure to H₂O₂ (**Fig. 3c**). Noted that the observation aligns with earlier pioneer research that utilized AOT to assist in the polymerization of ADPA and demonstrated the successful generation of polyaniline in its emeraldine salt form (PANI-ES) with absorption in NIR-II region (Please check Page 9).

REVIEWERS' COMMENTS

Reviewer #3 (Remarks to the Author):

Accept - their comments are fine.

Response to Reviewers' Comments

Reviewer #3 (Remarks to the Author):

Accept - their comments are fine.

Response: Thanks so much for your recommendation of publication.